# Fitness adaptations of Japanese encephalitis virus in pigs following vector-free serial passaging

Andrea Marti 1,2,3, Alexander Nater4, Jenny Pego Magalhaes1,2, Lea Almeida1,2, Marta Lewandowska1,2,3, Matthias Liniger1,2, Nicolas Ruggli1,2, Llorenç Grau-Roma2,5, Francisco Brito1,2, Fadi G. Alnaji6, Marco Vignuzzi6,7, Obdulio García-Nicolás1,2, Artur Summerfield 1,2*

1 Institute of Virology and Immunology IVI, Mittelhäusern, Switzerland, 2 Department of Infectious Diseases and Pathobiology, Vetsuisse Faculty, University of Bern, Bern, Switzerland, 3 Graduate School for Cellular and Biomedical Sciences, University of Bern, Bern, Switzerland, 4 Interfaculty Bioinformatics Unit (IBU) and Swiss Institute of Bioinformatics (SIB), University of Bern, Bern, Switzerland, 5 Institute of Animal Pathology, COMPATH, Department of Infectious Diseases and Pathobiology, Vetsuisse Faculty, University of Bern, Bern, Switzerland, 6 A*STAR Infectious Diseases Labs (A*STAR ID Labs), Agency for Science, Technology and Research (A*STAR), Singapore, Singapore, 7 Infectious Diseases Translational Research Programme, Department of Microbiology and Immunology, Yong Loo Lin School of Medicine, National University of Singapore, Singapore, Singapore

* artur.summerfield@unibe.ch

**Data Availability Statement:** All data acquired are in the main manuscript or the supplementary files. The sequence data is available through the European Nucleotide Archive PRJEB76411.

## Abstract

Japanese encephalitis virus (JEV) is a zoonotic mosquito-transmitted Flavivirus circulating in birds and pigs. In humans, JEV can cause severe viral encephalitis with high mortality. Considering that vector-free direct virus transmission was observed in experimentally infected pigs, JEV introduction into an immunologically naïve pig population could result in a series of direct transmissions disrupting the alternating host cycling between vertebrates and mosquitoes. To assess the potential consequences of such a realistic scenario, we passaged JEV ten times in pigs. This resulted in higher in vivo viral replication, increased shedding, and stronger innate immune responses in pigs. Nevertheless, the viral tissue tropism remained similar, and frequency of direct transmission was not enhanced. Next generation sequencing showed single nucleotide deviations in 10% of the genome during passaging. In total, 25 point mutations were selected to reach a frequency of at least 35% in one of the passages. From these, six mutations resulted in amino acid changes located in the precursor of membrane, the envelope, the non-structural 3 and the non-structural 5 proteins. In a competition experiment with two lines of passaging, the mutation M374L in the envelope protein and N275D in the non-structural protein 5 showed a fitness advantage in pigs. Altogether, the interruption of the alternating host cycle of JEV caused a prominent selection of viral quasispecies as well as selection of de novo mutations associated with fitness gains in pigs, albeit without enhancing direct transmission frequency.

**Funding:** This research was supported by Schweizerischer Nationalfonds zur Förderung der Wissenschaftlichen Forschung (https://data.snf.ch/grants/grant/192498) (192498 to AS). The funders had no role in study design, data collection and analysis, decision to publish, or preparation of the manuscript.

**Competing interests:** The authors have declared that no competing interests exist.

## Author summary

Japanese encephalitis virus (JEV) represents a major health threat in parts of Asia and Oceania. Primary vertebrate hosts are birds and pigs, but human infection also occurs and can cause severe encephalitis with high mortality. Like other Flaviviruses transmitted by insect bites, JEV requires replication in alternating cycles between mosquitoes on one side and birds or pigs on the other side. However, we previously reported that direct transmissions between pigs in absence of mosquitos can occur. Considering the increased risks for such events after the spread of JEV to a new region with immunologically naïve pigs, the present study was performed to understand if and how a series of direct transmissions would promote JEV adaptations to pigs and change virus-host interactions. Pigs infected with JEV passaged ten times showed enhanced clinical symptoms and stronger antiviral immune response, but luckily no increase in direct transmission was observed. Nevertheless, genomic analysis demonstrated a complete change in dominant virus variants, as well as selection of six viral amino acid changes. This indicates that interruptions of the alternating lifestyle of JEV causes a strong evolutionary pressure, which through fitness adaptations can change the viral characteristics.

## Introduction

Japanese encephalitis virus (JEV) is a zoonotic mosquito-borne Flavivirus endemic in temperate and tropical regions of eastern and southern Asia as well as Oceania. JEV is the most common cause of viral encephalitis in humans, with a mortality rate of up to 30% [1,2]. Survivors often suffer from neuropsychiatric sequelae [1–5]. The ecology of JEV is complex as it involves many vertebrate species and is characterized by cycling between mosquitoes vectors and certain vertebrates hosts requiring active replication in both vectors and hosts [6–8]. The main mosquito vector in Asia is *Culex tritaeniorhynchus* [9]. In vertebrates, a sufficiently high and long viremia is required to maintain the dual host cycling. Therefore, a prerequisite for this alternating change between vector and host is that Flaviviruses must be adapted to both, the insect and the vertebrate [10,11]. Main natural hosts of JEV are ardeid wading birds, but many different bird species develop viremia and produce an immune response with seroconversion [12,13]. Importantly, JEV also infects a variety of mammals, such as humans, horses, dogs, ruminants, and pigs [14]. Amongst those, only pigs have been identified to be relevant in the ecology of JEV because they are highly susceptible to JEV infection and also develop high levels of viremia. Thereby, pigs serve as amplifying host for JEV, which is particularly critical considering that pigs are often kept in high density and proximity to humans [15].

In light of the above, the observation that JEV-infected pigs efficiently shed the virus through their oro-nasal fluids and that direct transmission (DT) to naïve pigs occurs, represents a human and veterinary public health concern [16–18]. The impact of DT in the field is difficult to estimate, and likely to depend on many factors such as the level of immunity in the affected pig population and husbandry factors. Mathematical modelling using longitudinal data from pigs in Cambodia, in which JEV is endemic, supports a low rate of DT events between pigs in field conditions [19]. In this region, pigs were shown to be serologically naïve only during 2–6 months of age. Older and younger pigs were serologically positive either by previous infection, vaccination or maternal antibodies, and therefore likely resistant to infection [20,21]. However, this would be fundamentally different following JEV introduction into a new area with an immunologically naïve pig population. Such an event is not unrealistic considering the intense airway traffic, global trade and environmental changes introducing new

animal species and vectors into certain regions [22,23]. In fact, in 2021/2022 JEV spread to Australia, infecting piggeries and leading to 46 human encephalitis cases and seven deaths [24].

Given this considerable public health threat, the present study was initiated to investigate possible consequences following several DT events in pigs. We hypothesized that the evolutionary pressure caused by a series of DT in pigs, together with the high mutation rate of RNA viruses, will alter virological characteristics that could impact virus-host interactions and transmission potential. To this end, we investigated changes in clinical parameters in organ tropism, duration and magnitude of viremia, antiviral and inflammatory response, oro-nasal virus shedding, and transmission. Furthermore, next-generation sequencing was employed to analyze changes in viral populations and mutational adaptation occurring during a series of vector-free infections in pigs.

## Material and Methods (See S1 Text for additional details)

### Ethics statement

All experiments were performed at biosafety level (BSL) 3 and approved following the Containment Ordinance (ESV SR 814.912) by the Swiss Federal Office for Public Health and the Federal Office for the Environment (authorization number A110677-02). We also performed an internal risk-benefit evaluation (S1 Text). The experiments in pigs were conducted in compliance with the animal welfare regulation of Switzerland (TSchG SR 455; TSchV SR 455.1; TVV SR 455.163). The committee on animal experiments of the canton of Bern, Switzerland, reviewed the pig experimentation and pig blood collection protocols, and the cantonal veterinary authorities (Amt für Landwirtschaft und Natur LANAT, Veterinärdienst VetD, Bern, Switzerland) approved the animal experiments under the licenses BE101/19 and BE127/2020, respectively.

### Cells

*Aedes albopictus* C6/36 cells (ATCC) were cultured at 28°C and 5% $CO_2$. Vero cells (ATCC) and the porcine aortic endothelial cell line PEDSV.15 [25] (kindly provided by Dr. Seebach, University of Geneva, Switzerland) were cultured at 37°C and 5% $CO_2$. Porcine macrophages were derived from monocytes isolated from the blood of specific pathogen-free (SPF) Swiss Large White pigs as previously described [26]. Briefly, peripheral blood mononuclear cells were isolated by density centrifugation followed by monocytes purification using magnetic cell sorting. They were differentiated into macrophages and cultured at 39°C and 5% $CO_2$ [26]. Details of cell culture medium can be found in the S1 Text.

### Virus stock

A genotype I-b JEV strain, originally isolated from a human patient in Laos in 2009 (JEV_CN-S769_Laos_2009; kindly provided by Dr. Charrel, Aix-Marseille Université, Marseille, France) was used [27]. JEV stocks were produced using C6/36 cells infected with a multiplicity of infection (MOI) of 0.1 50% tissue culture infectious dose per cell ($TCID_{50}$/cell), determined by viral titration (S1 Text). After 72 hours post-infection (hpi) and culture at 28°C with 5% $CO_2$, the viral titers were determined by viral titration (S1 Text) as $TCID_{50}$ calculated according to the Reed-Muench formula [28,29]. The virus stocks were expanded three times on C6/36 cells before usage in this study as passage zero (P0).

### JEV Infection of pigs

Swiss Large White pigs from our SPF breeding facility were employed for all infection experiments. The piglets used for the experiments were 37–73 days old (S1 Table). To model how a

series of DT between pigs impacts JEV evolution, we performed a total of 10 passages (P) of JEV, employing a total of 30 pigs. The passaging was performed in three replicates. By keeping the three replicates separated, described as replicate line A, B and C, it enabled us to analyze the evolution of the virus over 10 passages in three different replicate lines. To start the passaging, three pigs were infected oro-nasally with $10^5$ TCID$_{50}$/animal of JEV P0 in 5 ml of MEM, with 1.5ml distributed in each nostril and 2ml in the mouth. Swabs and serum samples were taken at day 0 before infection and 3 and 4 days post-infection (dpi). The swabs were collected about 10cm deep and diluted directly after collection in 750µl of RA1 lysis buffer (Macherey Nagel) containing 1% β-Mercaptoethanol (Thermo Fisher Scientific) and frozen at -70˚C until further analysis. For the serum samples, blood was collected in 9ml S-Monovette neutral Z tubes (Sarstedt). After the collection the blood was allowed to clot at room temperature for about 1h before centrifugation at 2000 g for 15 min at 4˚C. The serum was collected and either used for a new infection or aliquoted and frozen at -70˚C for further analysis. For new infections, serum but not swabs were used, as this provided sufficient virus and volumes. To this end, 3ml of day 4 serum was inoculated intra-nasally, applying 1.5ml of serum to each nostril. The way of infection was switched from oro-nasal infections to intra-nasal infections as a lower volume of fluid was applied, using undiluted serum instead of the virus stock. The intra-nasal infection was selected, as the nasal epithelium was found to be highly susceptible *in vitro* [30] and based on previous experience [31]. This was performed for each of the three new pigs, keeping the replicate lines (A, B and C) separated. This was repeated until passage 10 (P10) was reached (Fig 1a). The clinical score (S1 Text) and the body temperature were measured daily. At 4 dpi of each passage, the pigs were euthanized by electrical stunning and exsanguination, and tissue samples from the mandibular lymph nodes, tonsils, thalamus, cortex and olfactory bulb were collected [32].

For the *in vivo* characterization of P10, two groups of five pigs were infected oro-nasally with 9 x $10^7$ viral genome copies/animal of either JEV P0 or P10 (corresponding to 1.8 x $10^5$ TCID$_{50}$/ animal for JEV P0). The latter represented a 1:1 mixture of P10 serum collected at 4 dpi from lines B and C. At 4 dpi, four naïve pigs were added to each group to evaluate DT rates. The infected pigs were euthanized at 11 dpi, while the in-contact pigs were kept until 15 dpi. Daily sampling of blood and swabs was performed as described in the first experiment. The evaluation of clinical scores and body temperatures was performed by veterinarians participating in the blind trail. At day 0 (before infection) and days 3, 7 and 11 post-infection, EDTA blood was collected. On the day of euthanasia, tissue samples were collected as described above.

## RT-qPCR

Viral RNA was extracted using the NucleoMag VET kit (Macherey Nagel) and the extraction robot Kingfisher Flex (Thermo Fisher). Viral RNA was quantified by RT-qPCR with the AgPath-ID One Step RT-PCR KIT (Thermo Fisher), using the forward primer 5'—ATC TGA CAA CGG AAG GTG GG– 3', the reverse primer 5'-TGG CCT GAC GTT GGT CTT TC—3' and the probe 5'–FAM—AGG TCC CTG CTC ACC GGA AGT–TAMRA -3'. Viral genomic RNA was quantified relative to a T7 *in vitro* transcribed reference JEV RNA (473 bases long sequence within the 3'-untranslated regions (UTRs) of JEV Laos), which was used as a standard (S1 Text For the standard curve, a dilution series spanning the range of $10^7$ to $10^1$ copies/µl was used. For the JEV Laos inoculum, $10^4$ copies/µl corresponded to 2 x $10^4$ U/ ml and 1U being defined as 1 TCID$_{50}$ [17]. The detection cut-off was defined at $10^1$ copies/ µl. Samples were analyzed using a 7500 Applied Biosystems real-time PCR machine (Thermo Fisher).

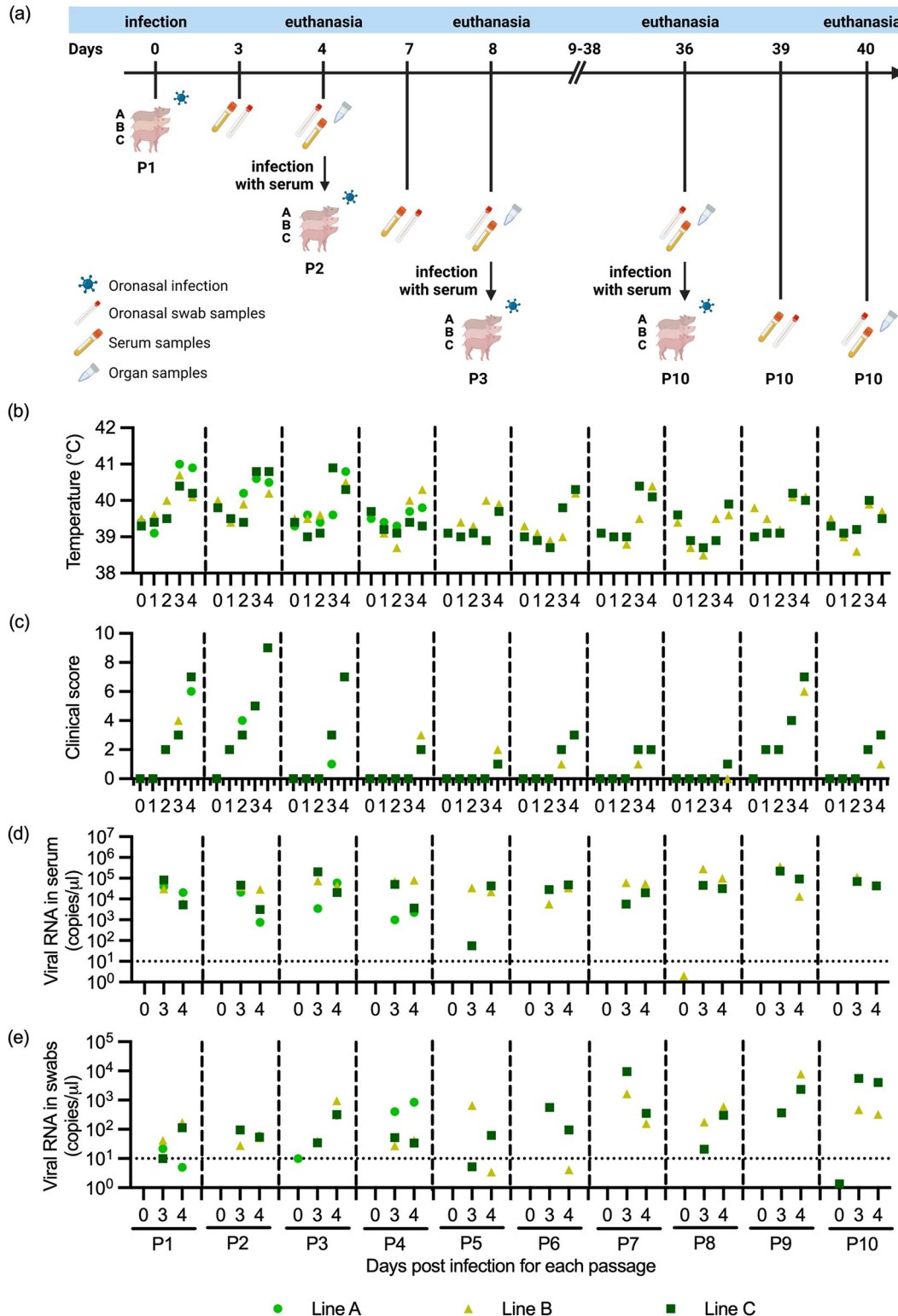

**Fig 1. Serial passaging of JEV in pigs.** In (a), the experimental layout and sampling time points are schematically represented (graphic created with BioRender.com). The passaging was performed in three independent lines A-C. For P1, three pigs were oro-nasally infected with JEV P0 (initial JEV stock, used as inoculum for the pigs infection). To generate P2-P10, pigs were oro-nasally infected with serum collected at four days post-infection of the previous passages. Line A was lost after P4. In (b)-(e), body temperatures, clinical scores, viral RNA loads in serum and in oro-nasal swabs are shown,

respectively. The clinical scores were determined following a clinical score sheet (S1 Text). Due to the loss of replicate line A, no statistical analysis was performed on the remaining two values.

## Transcriptomics

Transcriptomic analyses employed RNA extracted from blood leukocytes as described in the S1 Text. Libraries were prepared using the BRB-seq Library preparation kits (Alithea, Switzerland) at the Next Generation Sequencing (NGS) Platform of the University of Bern. Quality control employed a 5200 Fragment Analyzer CE instrument (Agilent) and sequencing the Illumina NovaSeq6000 sequencer. Reads were mapped to the pig genome (*Sus scrofa* 11.1, Ensembl release) using Tophat v.2.0.11 [33–36]. The number of reads overlapping with each gene was evaluated with Htseq-count v.0.6.1 [37,38]. The Bioconductor package DESeq2 v1.38.3 [39] was used to test for differential gene expression between the experimental groups. Gene set enrichment analysis (GSEA) was performed following ranking of genes based on differential gene expression using the "stat" value [40,41]. Calculations of normalized enrichment scores and false discovery rates (FDR) were performed using online tools available on https://www.gsea-msigdb.org [42]. Blood transcriptional modules (BTM) defined by Li et al. [43] for humans and modified for pigs [44] were used. Figures were created in R 4.3.0 using the ggplot2 package.

## Pathology

For tissue preparation and histopathological assessment please read the S1 Text.

## Serum neutralization assay

Serum neutralization assays were performed by adding serially diluted sera and 100 focus forming units of JEV per well using Vero cells. After incubation for 48h, the cells were stained for viral E protein to determine the 50% neutralizing dose ($ND_{50}$) of the sera. Details of the protocol are described in the S1 Text.

## Growth curves of passaged viruses

Confluent C6/36 and PEDSV.15 cells were infected with JEV isolated from sera at an MOI of 0.01 $TCID_{50}$/cell, determined by viral titration (S1 Text). After 1.5h of incubation, the inoculum was removed, the cells were washed twice with pre-warmed PBS, and fresh medium was added. The time point 0 hpi was harvested right after washing. The supernatant was further harvested at 18, 24, 48 and 72 hpi, and analyzed by RT-qPCR (see above), and by the determination of the $TCID_{50}$/ml by viral titrations on the mosquito C6/36 and the porcine PEDSV.15 endothelial cell lines (S1 Text) [28,29]. These cell types were selected to detect possible fitness changes of the passaged viruses in the porcine host and/or the insect vector.

## Cytokine quantification

Interferon gamma (IFN-γ), interleukin 1 alpha (IL-1α), IL-1β, IL-1ra, IL-2, IL-4, IL-6, IL-8, IL-10, IL-12, IL-18 and tumor necrosis factor (TNF) levels in serum samples of the infected pigs at -3, 3, 4, 5 and 7 dpi were quantified using the Milliplex MAP Porcine Cytokine/ Chemokine Magnetic bead kit (Millipore) and the BioPlex Magpix Reader (Bio-Rad). IFN-α was measured in duplicates by ELISA as previously described [45].

## Viral genome analyses

For the viral genome sequencing, RNA was extracted from serum or swab samples. For details on extraction, RNA quality and quantity evaluation, library preparations we refer to the S1 Text. Libraries were sequenced at 100 bp paired-end using an Illumina NovaSeq 6000 S4 Reagent Kit v1.5 (200 cycles; Illumina) on an Illumina NovaSeq 6000 instrument. The quality of the sequencing run was assessed using Illumina Sequencing Analysis Viewer (Illumina version 2.4.7) and all base call files were demultiplexed and converted into FASTQ files using Illumina bcl2fastq conversion software v2.20. Quality control of raw sequencing reads used FastQC v0.11.9 [46]. Removal of adapter sequences and extraction of unique molecular identifiers (UMIs) were performed with Fastp v0.23.4 [47]. We classified read pairs using Kraken2 v2.1.2 [48] by exact k-mer matching to a custom database consisting of multiple host and potential contaminant genomes, including GRCh38 (Homo sapiens), Sscrofa11.1 (*Sus scrofa*), GCF_006496715.1 (*Aedes albopictus*), and GCF_001876365.2 (C6/36), as well as JEV sequences KY927816, KC196115, and EF571853. We extracted all read pairs classified as being derived from JEV Laos for further analysis. The total number of reads, the viral reads and the covering depth for the viral genome of each sample are listed in S2 Table.

A *de novo* reference sequence of JEV Laos strain was assembled from pre-processed and filtered reads of the P0 sample using SPaDES v3.15.5 [49] in the 'rnaviral' mode. The resulting reference genome was manually curated and a GTF file of gene annotations was created based on alignments of protein-coding sequences to the assembled genome. Reads were mapped to the reference genome using BWA mem v0.7.17 [50] and SAM files were converted to BAM format with SAMtools v1.17 [51]. UMI sequences were moved from read names to a tag in the BAM files using the CopyUmiFromReadName command of fgbio v2.1.0 [52]. Run-wise BAM files were then merged for each sample and deduplicated using the MarkDuplicates tool of GATK v4.4.0.0 [53]. Based on the resulting duplicate-marked BAM files, single-nucleotide variants were called per sample using LoFreq v2.1.5 [54] without applying default filters for coverage or strand bias. Genome-wide mapping statistics were obtained with the flagstat command of SAMtools v1.17 [51] and Mosdepth v0.3.3 [55]. Analysis of the site-wise sequencing depth for each sample was performed with the depth command of SAMtools v1.17 [51]. Variant effects and sequence statistics were calculated with SNPGenie v1.0 [56]. Allele frequency trajectories and genetic diversity statistics along the passages were generated with custom Python scripts using the sample-wise VCF files and filtering for a minimum site-wise sequencing depth of 100 for serum and 10 for swabs, based on the output of SAMtools depth. Networks of pairwise allele frequency correlations between variant sites across passages were generated with Cytoscape v3.10.1 [57]. All statistical analyses were run in R version 4.2.1 (2022-06-23) [58].

## Statistical analysis

Statistical analyses of the non-bioinformatical datasets (S1 Data) were performed with Graph-Pad Prism 8.0 (GraphPad Software, La Jolla, USA). To compare differences at individual time points of the *in vitro* growth curves we used Tukey's multiple comparison test (ANOVA). Possible differences between the passages for the entire growth curves were tested using linear regression and Dunnett's multiple comparison test. Statistical significance for the clinical, virological and pathological data used the Mann-Whitney U test, comparing the P0 to P10 values at each time point. In addition, to treat time as a continuous variable, repeated measure ANOVA with Geisser-Greenhouse corrections were performed. To compare the areas under the curves for these data sets unpaired t-tests were performed. For the cytokines and neutralizing antibodies, we employed Mann-Whitney U tests. The significance levels were determined

by p values with *p<0.05, **p<0.01; ***p<0.001; **** p<0.0001. p values above 0.05 were considered non-significant.

## Results

### Serial passaging of JEV in pigs

To identify virus adaptations associated with a series of direct transmissions of JEV in pigs, a total of 10 passages were performed in three independent sets of pigs (replicate lines A-C) as depicted in Fig 1a. The passaging was performed blindly, not checking the viral loads in the animals. During passaging, the infectivity was lost at P5 in line A. Therefore, there is no line A data for P5-P10 in Fig 1. Most infected pigs had increased body temperatures and clinical scores at 3 and/or 4 dpi (Fig 1B and 1C). The piglets showed reduced liveliness and appetite during the peak of infection. All infected pigs were viremic at 3 and 4 dpi, with $10^4$ to $10^5$ viral copies/μl serum (Fig 1D). For the JEV Laos inoculum, $10^4$ viral copies/μl corresponded to $2x10^4$ TCID$_{50}$/ml. Viral RNA loads in the oro-nasal swabs were measurable at day 3 and 4 post infection, ranging from $10^1$ to $10^3$ viral copies/μl (Fig 1E). The viral RNA loads in central nervous and lymphoid tissues collected at 4 dpi are shown in S1 Fig. It is important to note that the virological and clinical differences between the passages in this animal experiment may also have been influenced by age effects (S1 Table) as well as by differences in the infectious dose. Furthermore, considering that from P5 on only two lines were left over, no statistical analysis was performed for this dataset. For these reasons, we further characterized the viruses isolated from P1-P10.

### Serum-isolated passaged JEV did not result in enhanced fitness *in vitro* in porcine cells

To identify possible changes in replication characteristics, we performed comparative growth curves in insect C6/36 and porcine PEDSV.15 cells. The analyzed viruses included JEV P0 (stock used to infect the first passage in pigs), P1, P5 and P10 stocks there were generated from serum obtained in the experiment described in Fig 1. To this end, it was necessary to rescue serum viruses with an antibody-dependent enhancement protocol to circumvent unknown serum components preventing virus rescue in some of the samples. Therefore, P1, P5 and P10 were rescued on macrophages, while the initial JEV P0 working stock derived from C6/36 cells.

Comparing the viruses in the growth curve on C6/36 cells, both lines of P1, P5 and P10 viruses demonstrated a clear delay in replication compared to P0, particularly visible at the 24 hpi. This observation was made at both viral RNA and progeny virus level, independently of the cell type used for titration (Fig 2A). To evaluate if this apparent loss of fitness in insect cells is also observed in porcine cells, we performed identical growth curves using PEDSV.15 cells (Fig 2B). The viral RNA loads were not significantly different between P0 and P1, P5 and P10, except for 18hpi. Nevertheless, P0 virus outgrew the passaged JEV in terms of viral titers determined in PEDSV.15 cells at 48 and 72 hpi. This was also observed for line C viruses when titrations used C6/36 cells. These data indicate that all viruses isolated from pigs had delayed and reduced *in vitro* replication characteristics compared to P0 JEV.

As all serum viruses were rescued on macrophages, while the P0 working stock came from C6/36, the measured growth kinetics were additionally statistically analyzed, comparing P1 with P5 and P10. This analysis identified enhanced fitness of P10 (and sometimes also P5) in line C when grown on C6/36 cells. At 48–72 hpi higher titers were observed independent of the cell type used for titration (Fig 2A). This enhanced fitness was not observed with the growth curves in PEDSV.15 cells (Fig 2B) and neither in the growth curves of line B.

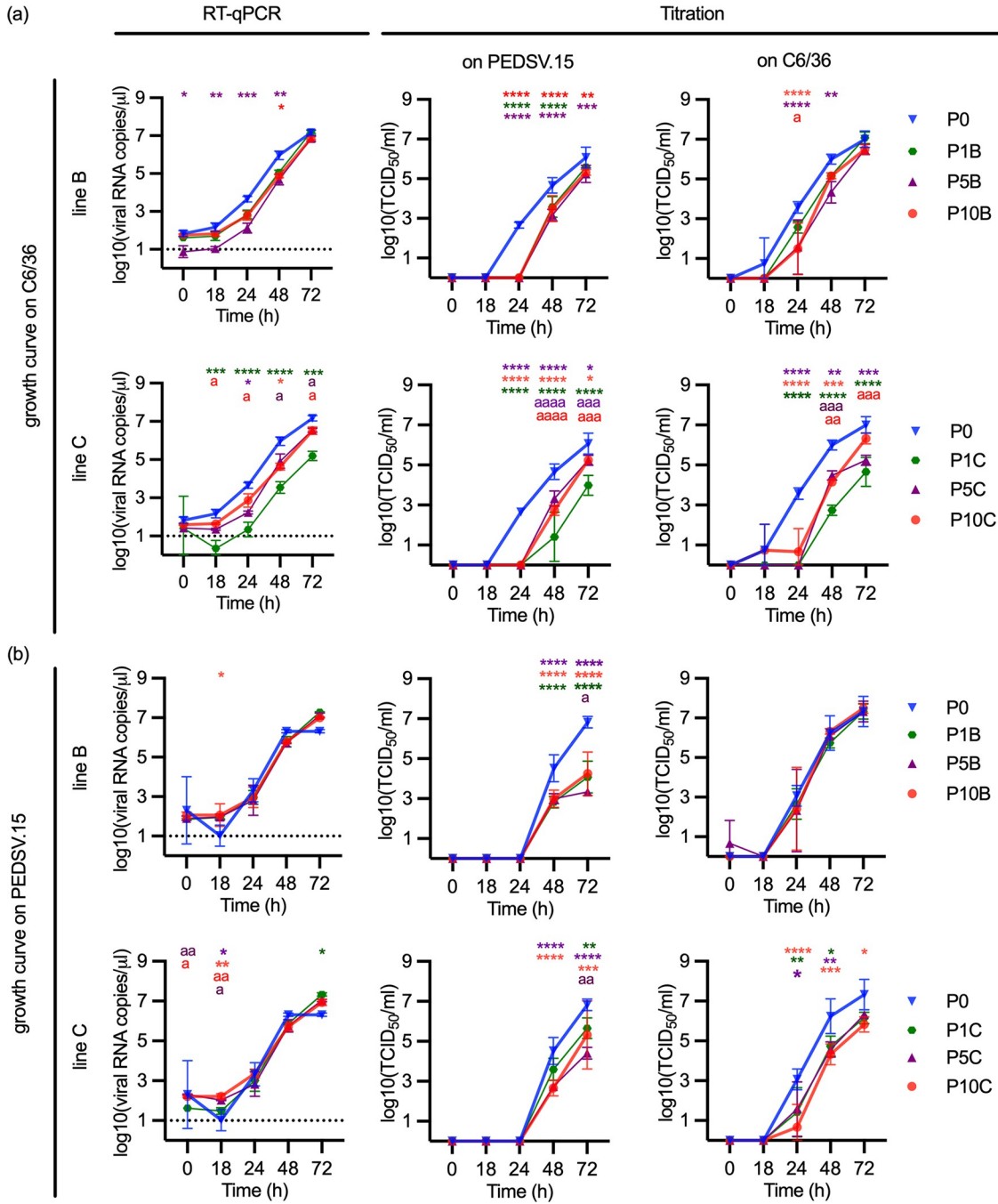

**Fig 2. *In vitro* growth curves of passaged JEV isolated from pig serum.** The insect cells C6/36 (a) or the porcine cell line PEDSV.15 (b) were infected with a MOI of 0.01 $TCID_{50}$/cell with the initial JEV stock P0 (blue) and serum-isolated JEV of P1 (green), P5 (purple) or P10 (red) in triplicates. Passage lines B and C were depicted separately. The presence of the virus was analyzed by RT-qPCR, as well as by titration on C6/36 and PEDSV.15 cells. The use of both cell types for titration enables to detect possible fitness advantages or disadvantages of the passaged viruses in the porcine or insect cell lines, as adaptation might generate not only a gain of fitness in the respective species but can as well cause a loss of fitness in the other species. Statistical significance was determined using Tukey's multiple comparison test (ANOVA) using either the P0 values as reference (*$p < 0.05$, **$p < 0.01$; ***$p < 0.001$; ****$p < 0.0001$) or following exclusion of P0 using the P1 values as reference ([a]$p < 0.05$, [aa]$p < 0.01$; [aaa]$p < 0.001$; [aaaa] $p < 0.0001$). The same color code as above was used to indicate the affiliated groups.

The viral growth curves were also analyzed by linear regression and Dunnett's multiple comparison test (S3 Table). This confirmed significant reductions in growth characteristics of the serum-derived viruses from line B when compared to P0, if these viruses were grown and titrated on PEDSV.15. In addition, P1C and P5C showed significantly a decreased growth compared to the P0 virus when grown on PEDSV.15 and titrated on PEDSV.15 and C6/36, respectively. When these viruses were grown and titrated on C6/36 we also observed reduced growth characteristics with P1, P5 and P10 when compared to P0. On the other hand, when comparing P1, P5 and P10 only, no significant differences between them were found (S3 Table). These results indicate that the main *in vitro* differences observed in these tests, related to the cell culture history and isolation procedure, rather than the effects of passaging.

## Passaged JEV increased viremia, nasal shedding and clinical symptoms

To identify if a series of direct transmissions modelled by passaging the virus in pigs could result in possible changes in virulence, organ tropism, virus shedding and direct transmission capacities, we infected five pigs with either JEV Laos P0 or a 1:1 mix of pig serum P10B and P10C (Fig 3A). The P0 infected groups are stated as Passage 1* (P1*), to differentiate from the passage 1 of the previous animal experiment, while the P10 infected pigs are termed as passage 11 (P11). For both groups, the clinical, virological, pathological and immunological parameters were assessed (Figs 3, 4 and 5). In addition, at 4 dpi four naïve pigs were co-housed to obtain information on possible changes in direct transmissibility (Fig 6). The infected pigs were euthanized at 11 dpi, while the in-contact pigs were kept until 15 dpi.

When comparing the P11 to P1*, the P11 pigs showed a similar course of disease, but with signs of enhanced virulence (Fig 3B–3G). With P11 we observed a significantly increased body temperature at 3, 4 and 8 dpi (Fig 3B), higher clinical scores at 3 and 7 dpi (Fig 3C), higher viremia at 3 dpi (Fig 3D), higher viral RNA loads in nasal swabs at 8 dpi (Fig 3E) and higher viral loads in the thalamus at 11 dpi (Fig 3G). The enhanced clinical scores and nasal shedding were confirmed by area under the curve analyses (Fig 3C and 3E).

After euthanasia, JEV RNA was identified in the cortex, thalamus, olfactory bulb, tonsils and mandibular lymph nodes with the highest levels in the tonsils (Fig 3G). This was similar between P1* and P11 groups and was consistent with previous observations [17,32,59]. All infected pigs showed mild to moderate histopathological lesions in the CNS without group differences in the infected pigs (Figs 3H and S2) indicating that organ tropism and neuro-invasiveness of JEV was not affected by the passaging. No lesions were observed in the tonsils and mandibular lymph nodes.

## Passaged JEV stimulated stronger innate immune responses

Considering that no information on *in vivo* cytokine responses in pigs were available, and that a series of vector-free direct transmission in pigs could also impact how JEV interacts with the host immune system, we investigated the innate and adaptive immune responses in serum samples from the experiment described in Fig 3A. Following infection with both viruses, increased levels of IL-12, IL-6, IFN-α and IL-1RA were found between 3–7 dpi (Fig 4A–4D). Other cytokines, including IL-1α, IL-2, IL-10 and IL-18, were significantly elevated only at individual days and not with both viruses (Fig 4E–4H). GM-CSF, IL-4, IL-1β, TNFα and IL-8 were not induced systemically by JEV (S3 Fig). Altogether, this cytokine profile indicates that JEV induces a Th1 and antiviral response in which proinflammatory responses are well-controlled. When comparing the P1* and P11 groups, we found increased early IFN-α and anti-inflammatory IL-1RA responses in the P11 pigs (Fig 4C and 4D).

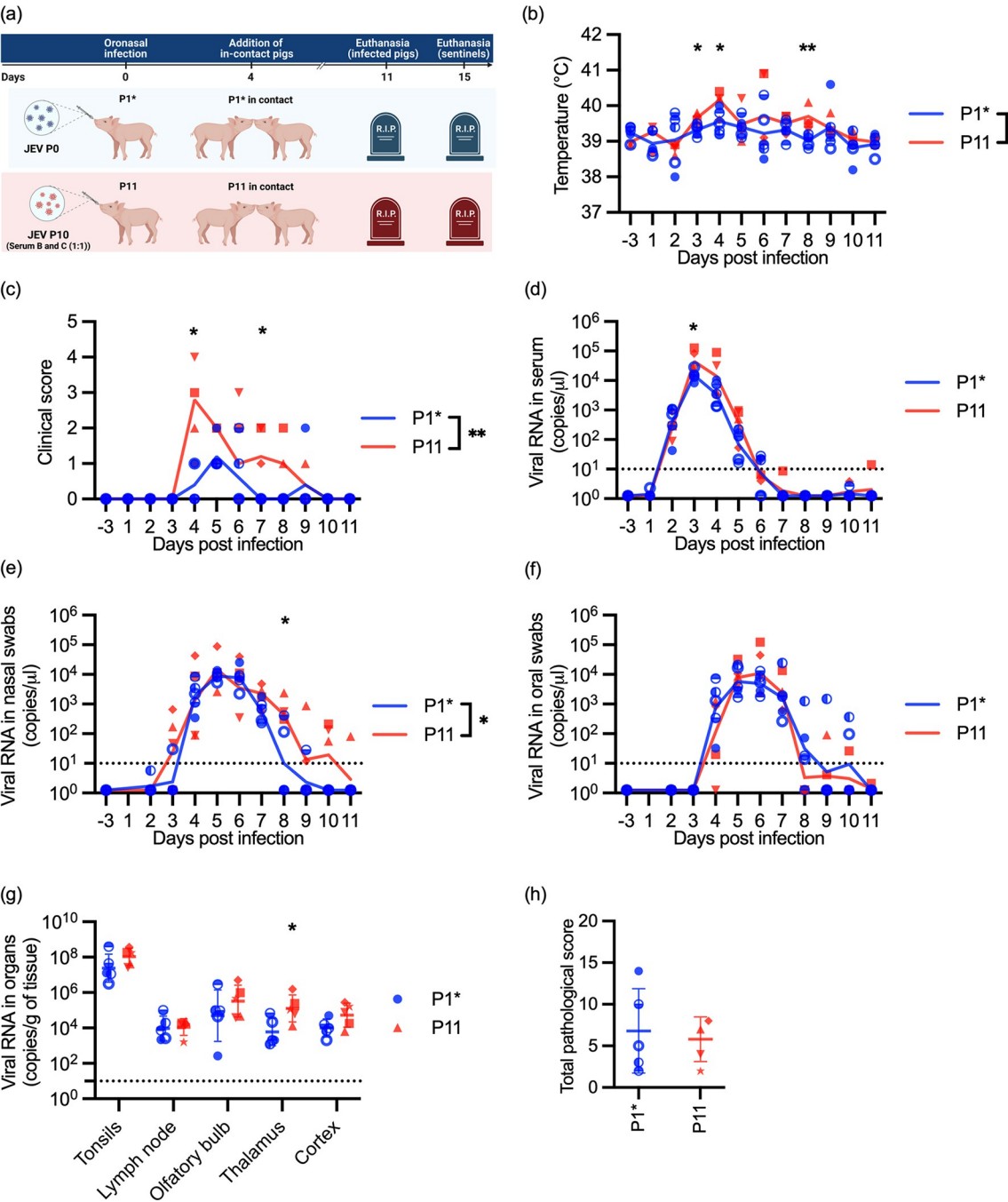

**Fig 3. In vivo characterization of passaged JEV–clinical, virological and pathological data.** In (a) a schematic representation of the animal experiment is shown (created with BioRender.com). Five pigs were oro-nasally infected with JEV P0 and five pigs with P10. The pigs infected with the initial JEV stock P0 were labelled as P1* to differentiate from the first animal experiment, while the group infected with the mixture of P10 serum was termed P11. At 4 dpi, four naïve non-infected pigs were added to each group to determine direct transmission event. In (b)-(h), data from the oro-nasally infected pigs is shown, including body temperature (b), clinical scores (c) serum viral RNA loads (d), viral RNA loads in nasal swabs (e), viral RNA loads in oral swabs (f), viral RNA loads in CNS and lymphoid tissue (g) and histopathological scores of CNS tissues (h). This score represents additive data from 10 brain tissue samples (details in S2 Fig). Statistical analyses used the Mann-Whitney U test, comparing the P0 to P10 values at each time point. In addition, a repeated measure ANOVA with Geisser-Greenhouse correction was performed in (b) to (f). The significance is indicated in the legends of the panels. Comparing the areas under the curves for the two groups in Figures (c)-(f) by an unpaired t-test, the area under the curve results in equal significance as the area under the curve identified in the repeated measure ANOVA. The significance levels for all statistical tests are indicated as *p<0.05, **p<0.01; ***p<0.001; ****<0.0001.

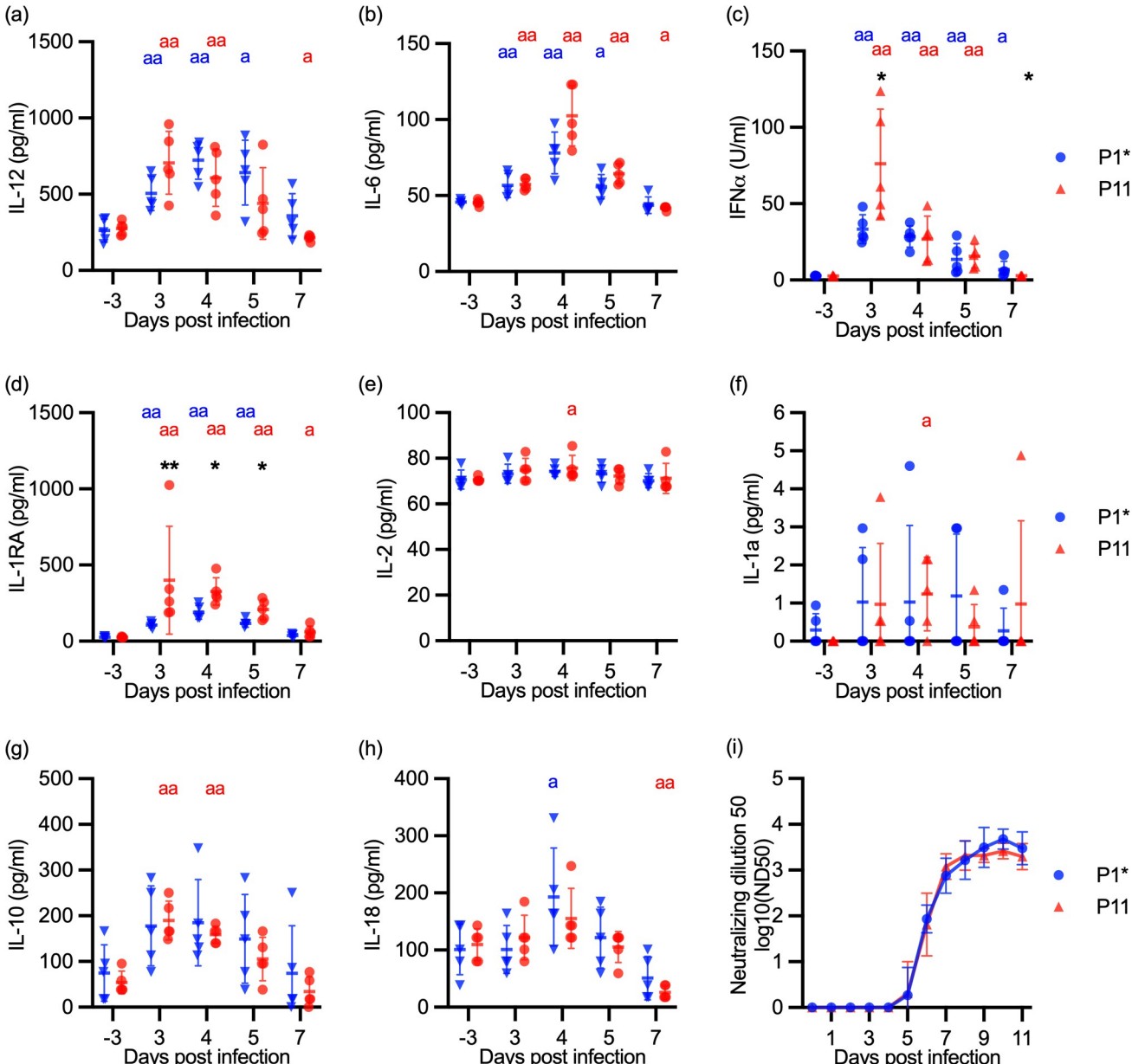

**Fig 4. Cytokines and neutralizing antibodies in the serum of JEV-infected pigs.** Sera from the animal experiment schematically represented in Fig 3A was analyzed. Panels (a-h) show the levels of IL-12, IL-6, IFN-α, IL-1RA, IL-2, IL-1a, IL-10 and IL-18, respectively. Panel (i) displays the levels of neutralizing. The statistically significant differences between groups on the same day are depicted as an asterisk, whereas differences compared to day -3 within the same group is depicted by the letter "a". All statistical analyses were done using the Mann-Whitney U test ([a]/*p<0.05, [aa]/**p<0.01; [aaa]/***p<0.001; [aaaa]/****<0.0001).

Independently of the infected group, antibody neutralizing titers were detected as early as 5 dpi, reaching very high titers at 9 dpi (Fig 4I). This kinetic appeared to coincide with control of viremia (see Fig 3D).

To further elaborate on innate immune responses induced by JEV, and on how passaging would impact such responses, we performed a transcriptome analysis of blood leukocytes before infection and at 3, 7 and 11 dpi from the P1* and P11 groups. We employed GSEA analyses with porcine BTM as gene modules, which have been demonstrated to provide

comprehensive immunological information following virus infection [44,60,61]. BTM related to the innate immune system were grouped in antiviral, dendritic cell (DC), inflammation, myeloid and NK cell BTM families (Fig 5A and 5B). In both groups, JEV induced the expected antiviral response at 3 dpi, which was followed by an enhancement of the NK cell BTM at 7 and 11 dpi. While JEV infection decreased expression of many DC, inflammation and myeloid cells at three days post-infection in the P1* group, this was not observed for P11, which increased expression of a few of these BTM such as M165, M67, S11 (all activated DC), as well as M86.0, M27.0 (chemokines and inflammatory mediators). Nevertheless, at 7 and 11 dpi, both groups showed a similar prominent downregulation of DC, inflammation and myeloid cell BTM, possibly related to the anti-inflammatory cytokine responses (Fig 4).

To investigate group differences in more detail, the transcriptome profiling on each day was compared between the groups (Fig 5B). This clearly confirmed that early innate immune responses were stronger following infection in the P11 pigs. More specifically, seven antiviral BTM, 11 BTM related to DC, 16 BTM related to inflammation, 13 myeloid cell BTM and one NK cell BTM were higher in the P11 compared to the P1* group. In a later stage of the infection (7 dpi) immunoregulatory mechanisms leading to a downregulation of certain of the innate BTM were more pronounced with P11 compared to P1* (Fig 5B).

We next investigated BTM related to the adaptive immune system. These were further classified as "B-cells", "cell cycle", and "T-cells" BTM (Fig 5C and 5D). Overall, the pattern of BTM induced in the two groups looked similar (Fig 5C). Both viruses induced a plasma cell response (M156.1) at 7 dpi, whereas all other BTM were downregulated. In contrast, cell cycle BTM were mostly induced, in particular at 7 dpi. Similarly, T cell BTM were also strongly induced at all days for the P1* pigs and at 7 and 11 dpi for the P11 pigs (Fig 5C). Comparison of the groups at individual dpi did not reveal many differences in the BTM expressions but confirmed a higher T-cell activation in the blood of the P1* (Fig 5D).

Taking together, JEV induced an antiviral and Th1 immune response accompanied by a potent neutralizing antibody response and an anti-inflammatory regulation. The higher innate immune responses found after infection with passaged JEV are possibly a consequence of increased virus replication rates. Our data indicate also that there was no acquisition of improved immune evasion capacities because of passaging.

## Passaged JEV showed no increased vector-free transmissibility

To determine possible changes in direct transmission, four in-contact animals were added to each group of the oro-nasally infected pigs. As shown in Fig 6, only one in-contact sentinel pig was infected in the P1* group. In this animal, the kinetics of JEV in serum and swab samples were similar to that seen with oro-nasally infected animals (Fig 6A and 6B). This pig was also the only sentinel pig that developed neutralizing antibodies (Fig 6C) and showed histological lesions in the brain (Fig 6D).

## 25 single-nucleotide deviations were positively selected

Taking all P0-P10 viruses isolated from the serum into consideration, single-nucleotide variations in 10% of the JEV genome locations were observed (Fig 7A). In the inoculum P0, 220 nucleotide deviations from the reference sequence of JEV Laos (GenBank: KC196115.1) were found. Taking together P1 to P10, additional 8% of the nucleotide positions expressed SNVs compared to the inoculum P0. The nucleotide deviations were distributed throughout the viral genome, with being 33% synonymous and 58% non-synonymous changes. 2% of the mutations led to stop codons and 8% were located in the UTRs (Fig 7A and S4 Table). To focus on positively selected nucleotide deviations, we filtered for a frequency above 35% in at least one

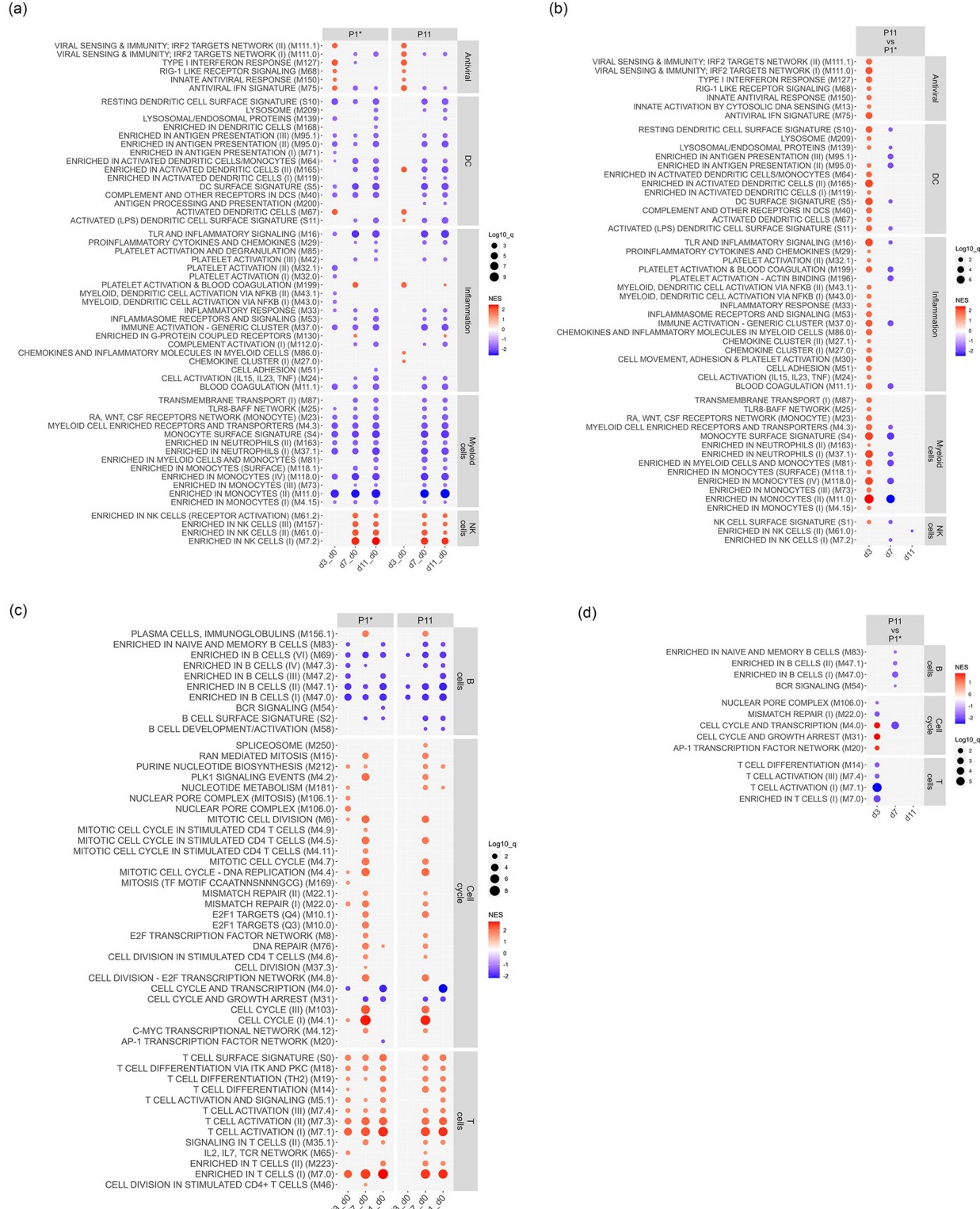

**Fig 5. Transcriptomic profiles leukocytes of the P1\* and P11 groups.** Blood leukocytes from uninfected, and JEV-infected at 3, 7 and 11 dpi of the P0/P10 *in vivo* characterization experiment (Fig 3A) were subjected to mRNA sequencing and analyzed by GSEA using porcine BTM gene sets. The dot plots show the normalized enrichment scores (NES; red upregulated, blue downregulated) with q-values indicated by dot sizes (cut-off at 0.05). The innate BTM gene sets (left y-axis labels) shown in (a) and (b) were further classified as "antiviral", "DC", "inflammation", myeloid cells", and "NK cells" (right y-axis label). The adaptive BTM shown in (c) and (d) were classified as "B cells", "cell cycle", and "T cells". In

(a, c), the timepoints of infected pigs (n = 5) were compared to uninfected pigs ("d0"), using the same baseline for the two groups. In (b, d), a comparison of JEV P10 (P11 group) versus P0 (P1* group) on different dpi is shown.

line and passage. This cut-off was chosen to focus on high frequency selected SNV and include those that emerged late during passages (see next chapter). This identified 25 single-nucleotide deviations, three in the UTRs, 16 synonymous and six non-synonymous. From the latter, two amino acid changes were in the envelope (E), two in the non-structural protein 5 (NS5), one in the precursor of the membrane (prM) and one in the non-structural protein 3 (NS3).

To determine changes in genetic variability, we calculated the nucleotide diversity ($\pi$) for each sample of the three lines. $\pi$ represents the average number of nucleotide differences per site between two viral haplotypes randomly chosen from the viral quasispecies populations. As

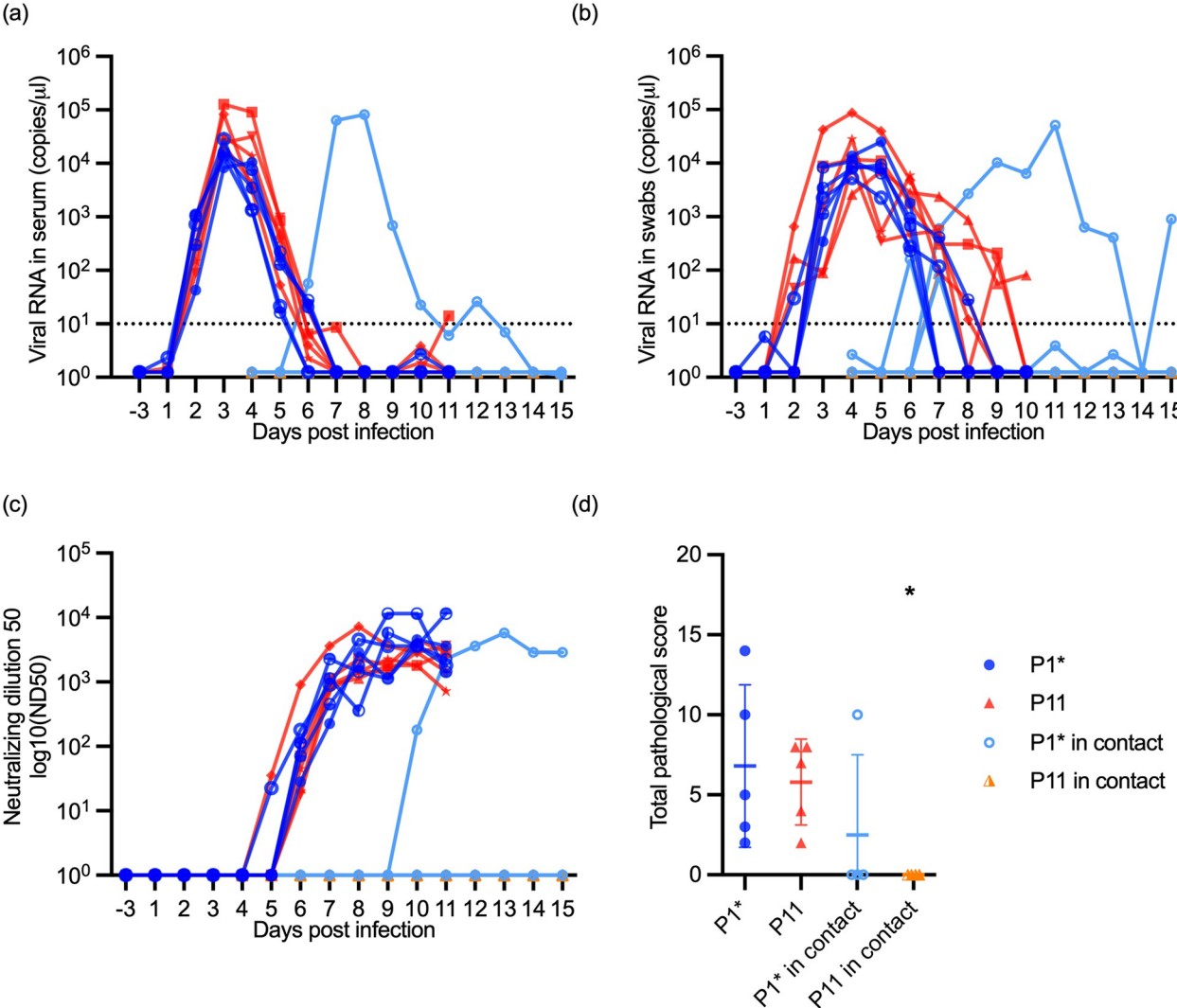

**Fig 6. Direct transmission of JEV between pigs.** Five pigs were oro-nasally infected with initial JEV stock P0 (P1* group) and five pigs with the *in vivo* passaged JEV P10 (P11 group). At four days post-infection, four pigs were added to each group to determine a possible direct transmission. In (a) and (b), viral RNA loads in the serum and nasal swabs are shown, respectively. Only one of the total eight sentinels (light blue line, in contact with the P1* pigs) got viremic (a) and positive for viral RNA in oro-nasal swabs (b). In (c), the neutralizing antibody titers are shown, and in (d), the pathological lesion scores in the CNS.

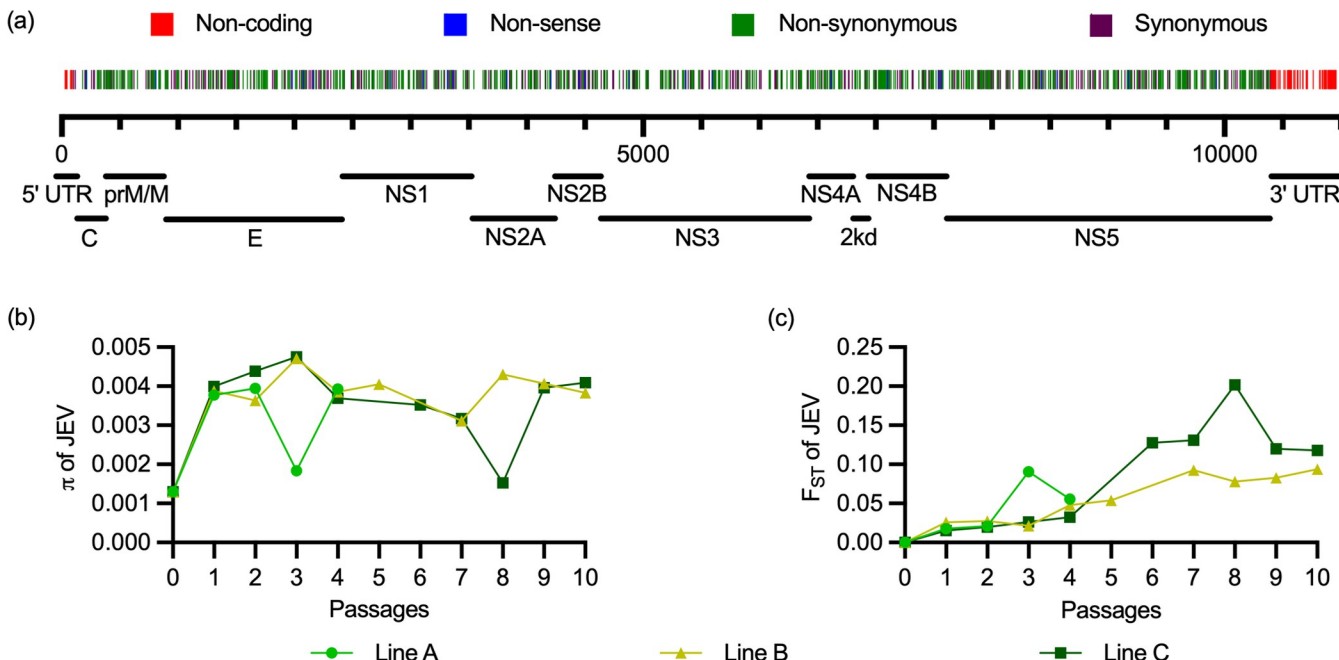

**Fig 7. Genomic changes during passaging.** The viral genome of JEV was extracted from serum samples of infected pigs at 3 dpi and analyzed by NGS. In (a), the single-nucleotide variants found in all passages across the genome are shown. Mutations in the UTR regions are depicted in red; those leading to stop codons in blue (non-sense), non-synonymous mutations in green and synonymous mutations in purple. In (b), the nucleotide diversity π calculated across the whole genome is shown. In (c), the pairwise genetic differentiation between viral populations in the initial JEV stock P0 and each passage is shown as fixation index $F_{ST}$, wherein 0 means no genetic differentiation and 1 complete genetic differentiation. In (b) and (c), each line of passaging is shown separately.

shown in Fig 7B, π increased from P0 to P1 possibly pointing on an initial diversifying selection at the expense of dominant populations present in P0. Between P1 and P10 π remained at a stable level indicating a balance between mutational diversification and selection processes.

As a measure of differentiation of viral populations along the passaging, we calculated the Fixation Index $F_{ST}$, based on the pairwise variance in allele frequencies between P0 and the passaged viral populations. Zero indicates no genetic differentiation between P0 and the passage of interest, while 1 indicates complete differentiation. As expected, an increase of $F_{ST}$ over time was observable for all three lines during passaging (Fig 7C). Temporary increases in $F_{ST}$ were observed in lines A (P3) and line C (P8) and associated with reduced π values (Fig 7B and 7C). The data also indicates that in line C, more prominent population differentiations were observed compared to line B. We also calculated the π and the $F_{ST}$ for each protein and the UTRs separately (S4 Fig and S5 Table). As expected, the π profiles of individual viral genes were mostly comparable to those of the whole genomes. One exception was the NS4B, which had already a high diversity in P0 that was not further enhanced during passaging. For $F_{ST}$, most viral genes had unique profiles indicating selection of mutations present on individual genes.

## Trajectory analyses identify JEV variants with fitness gain in pigs

We next made trajectory analyses for the 25 single-nucleotide variants found in the serum. Some of these single-nucleotide deviations followed quasi-identical trajectories along the passages indicating their localization on the same viral RNA molecule (Fig 8A). Based on this feature, we defined viral haplotypes when the frequencies of at least two nucleotide deviations were quasi-identical over the passages (Cosine similarity > 0.999 between the frequencies

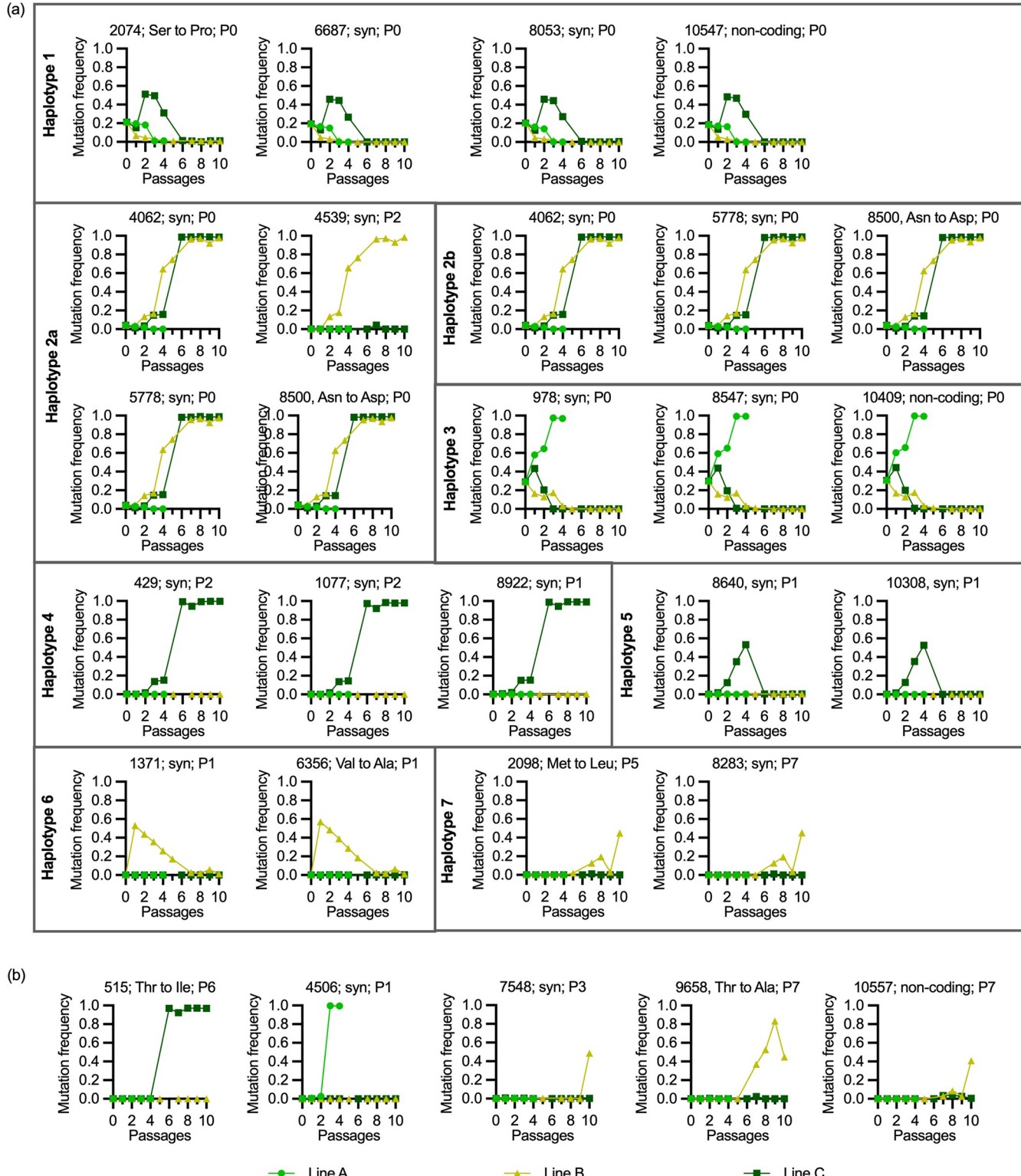

**Fig 8. Trajectory analyses of viral haplotypes and mutations.** Viral RNA was extracted from the serum on 3 dpi and analyzed by next-generation sequencing. Only mutations were considered where an allele different from the major allele in the initial JEV stock P0 reached a frequency of at least 35% in one of the passages. In (a), the mutations were grouped in different haplotypes that followed quasi-identical allele frequency trajectories along the passages. These were defined when a cosine similarity between the trajectories of nucleotide mutations in the haplotype was > 0.999 for any comparison in that haplotype. In (b), individual mutations that did not cluster in haplotypes are shown. For all plots, the position of the mutation, its impact on the protein sequence with syn standing for synonymous mutation and the passage number of the first detection of the mutation is shown.

trajectories of single-nucleotide deviations in the haplotype). Of note, due to the short-read sequencing, a final proof for haplotypes is missing. However, the very similar frequencies suggest that these haplotypes were most likely co-inherited and therefore present on a single viral genome.

Haplotypes 1 and 3 were present at high frequency in P0 (20% and 28%, respectively) but were de-selected in the surviving lines after 2–5 passages. In line A, haplotype 3 reached >97% frequency by P3, before this line became extinct after P4 (Fig 8A). Three mutations from haplotype 2b (4062, 5778 and 8500) were also detectable in P0 at frequencies of around 3%, and were selected to over 95% in lines B and C. In haplotype 2a, a mutation at position 4539 evolved at P2 in line B only. The remaining haplotypes 4–7 were not detectable in P0 and we cannot determine if these were selected from pre-existing minor variants in P0 or evolved by simultaneous mutations on one strand of the viral genome. Haplotype 4 only arose in line C and reached >95% frequency. Haplotypes 5 and 6 arose in P1 of only one line (C and B, respectively) and disappeared after initial selection. Finally, haplotype 7 was detectable only after 5–7 passages in line B, indicating that it probably emerged by simultaneous mutations on two sites of one genome. In addition to the above haplotypes, a total of five individual mutations were selected to reach levels over 35% (Fig 8B and S4 Table).

## Competitive advantage for haplotype 2, haplotype 7 and mutations at positions 4539, 7548 and 10557

To determine fitness differences in haplotypes and mutations present in lines B and C, we also sequenced the serum viruses obtained from the comparative P0/P10 pig infection experiment, in which the P10 inoculum represented a 1:1 mixture of lines B and C. These viruses obtained from a total of five pigs were termed P11. To calculate the mutational frequencies in the P10B/C inoculum the average frequencies of the SNV known to be in the original P10 was calculated. This was then compared to the frequencies in the P11. The only single-nucleotide variants staying at 100% were those of haplotype 2b. Haplotype 7 (2098 and 8283 mutations in line B) showed a relative increase, while haplotype 4 (mutations 429, 1077 and 8922 in line C) showed a relative decrease in frequencies (Fig 9A and S4 Table). A positive selection was also observed for mutations 4539, 7548 and 10557 emerging in line B. Only two emerging mutations that caused amino acid changes showed a fitness advantage in this experiment. One was in haplotype 7 at position 2098 encoding the E protein (Met→Leu), and the other in haplotype 2 at position 8500 encoding the NS5 (Asn→Asp). Of note, the animal in which these line B haplotype/mutations were mostly selected also had the highest viremia and IFN-α levels (square symbol in Figs 3, 4 and 9).

To determine possible differences in variant selection we also sequenced the swab samples. The number of reads of these samples was often too low to provide reliable data on the frequencies but for those that were of sufficient depth, the data indicate similar frequencies for the dominant SNV that were found in the serum (S6 Table). This is best visible for the swab sample of pig 2029 (which hat the highest coverage of all; S7 Table). This data does not point on a bottle neck effect for JEV to reach the oronasal mucosa for shedding.

## Passaging of JEV in pigs results in prominent selections of pre-existing variants to change the overall quasispecies composition

Fig 9B visualizes the evolution of divergent nucleotides that were already present in P0. Many of these variants were still present at comparable frequencies in the serum derived viruses from P1* but not from P11. In all pigs, the dominant quasispecies had dramatically changed. Dominant variants with frequencies >5% in P0 were mostly undetectable in P11 at the

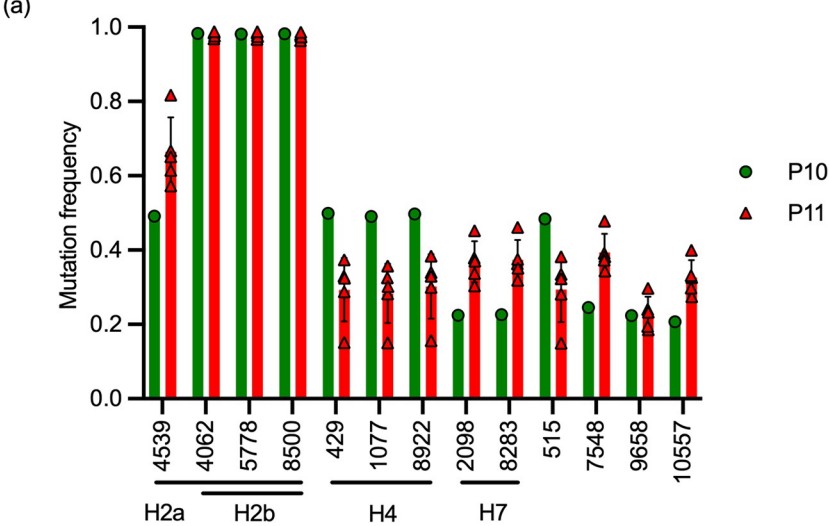

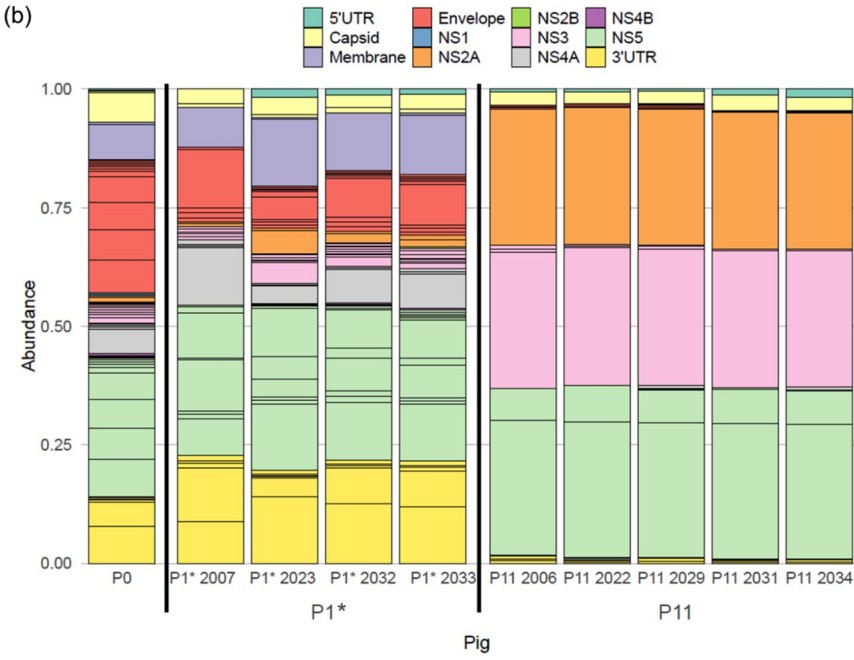

**Fig 9. Selection of haplotypes and divergent nucleotides during passaging.** JEV was passaged *in vivo* 10 times, resulting in P10 infectious serum. To check possible viral adaptations the P10 serum of lines B and C was used to further infect five pigs, resulting in P11. P11 was compared to a group of five pigs, infected with the initial JEV stock P0 (resulting in P1*). Serum samples were collected at 3 dpi for next-generation sequencing analyses. (a) Haplotypes and divergent nucleotides frequencies of input (P10, green) and output JEV (P11, red) are shown. For P10 the average frequencies of line P10B and P10C was calculated. For P11 each pig was depicted separately, with n = 5 (symbols), while the bars represented the average frequencies. Only nucleotide changes reaching at least 35% in frequency are shown. (b) All nucleotide positions of the initial JEV stock P0 divergent from the JEV Laos reference genome were selected and the frequencies in P0, P1* and in P11 were plotted to visualize major positive and negative selections.

expense of minor variant amplification (Fig 9B and S4 Table). This indicates that a series of direct transmission events in pigs is associated with strong selection and de-selection processes of pre-existing minor variants.

## Discussion

Mosquito-borne Flaviviruses have evolved by adaptation to both insects and vertebrates to maintain an alternating host cycle. The clear genetic, physiological and immunological differences between insects and vertebrates requires viral adaptation, processes that may involve fitness trade-offs in both or in one of the hosts [62]. The high mutational frequency of Flaviviruses in the range of $10^{-3}$ to $10^{-5}$ per replicated nucleotide creates a swarm of mutants which enables the selection of pre-existing quasispecies, as well as the continued selection of mutations to ensure fitness in the current host [63]. It should be noted that trade-offs following adaptation to one host have not always been observed with Flaviviruses [64]. Furthermore, despite the relatively wide host plasticity of many vector-borne viruses, Flaviviruses have adapted to efficiently replicate in rather selective vertebrate hosts, for JEV being birds and pigs.

Given the ability of JEV to transmit directly between pigs in contact [17], a series of vector-free transmission events would be possible after introduction of JEV into a herd of immunologically naïve pigs. Hence, the present study addressed whether and how such an event could alter virus pathogenesis and viral genetic features. Our data demonstrate that JEV passaging in pigs induced virus adaptations associated with enhanced viremia, nasal shedding, clinical scores and innate immune responses. This can be interpreted as an increase in viral fitness and relates to previous observations found with other Flaviviruses following passaging in vertebrates such as West Nile virus (WNV) in chicks [65], Zika virus (ZIKV) in mice [66–68] and also JEV in mice [69]. However, JEV passaging in suckling mice was also reported to result in attenuation [70]. Nevertheless, considering that mice are not a natural host of ZIKV or JEV, the evolutionary pressure on the virus during experimental passaging is expected to be quite different.

The higher viremia and the prolonged shedding indicate a viral fitness gain for the passaged virus in pigs. On one side, the stronger innate immune responses which was demonstrated both in terms of serum cytokines and leukocyte transcriptome might be a direct consequence of these higher levels. On the other side, the data indicates that the virus did not gain an increased capacity to escape the innate immune responses.

The levels of parameters for the adaptive immune responses, was similar when comparing P1* to P11. The similar antibody responses were confirmed by the transcriptomic response showing a plasma cell response at day 7 in both groups. The cell cycle response typically seen during the clonal expansion of lymphocytes was also prominent at day 7 for both groups. Finally, the transcriptomic analyses also support a potent T cell activation in both groups, although in this case, the onset was already seen at day 3 in the P1* group. Altogether, the data do not indicate that JEV has acquired immune evasion capabilities during passaging.

The importance of the T-cell responses against JEV was also observed for humans [71] and mice [72]. The induction of a Th1 response by JEV was previously also described for mice [73,74] and humans [71,75,76]. However, mice experiments showed some differences in the T cell response regarding the route of inoculation. While the intraperitoneal and subcutaneous infection of mice induced a high Th1 response, the peroral route induced for some experimental conditions as well a Th2 response [77]. Induction of IL-4 after infection of the piglets with JEV was not observable for our analyzed time points. If an induction of IL-17$^+$ CD4$^+$ T cells is observable in pigs, as described by Eun-Young Lee *et al.* [78] for the sublingual administration in mice remains unknown as we did not verify the cytokine level of IL-17 or T cell population

in the blood. Nevertheless, our transcriptomic and cytokine data do not support the presence of a prominent inflammatory response.

We were unable to find evidence for an increased direct transmission rate following passaging, despite the fitness gain observed by the passaged virus. This was unexpected considering that the longer nasal shedding should favor direct transmission. In fact, viral doses as low as 10 $TCID_{50}$ applied via the oro-nasal route to pigs were found to produce a JEV infection [17]. A possible explanation could be that the stronger clinical symptoms in the P10-infected pigs reduced the contact between animals. Our transmission data combined with our previous observations confirms the possibility of direct transmission between pigs, but also indicate that these may be relatively rare events requiring intensive contact between pigs. Nevertheless, it should be noted that our pigs were kept at a low stock density (>3 m$^2$ per animal). Commercial farming uses 0.7–1 m$^2$ of space per pig, depending on the country [79–81]. It is conceivable that under dense commercial farming conditions direct transmission events could be favored [82,83]. It is also possible that direct transmission is influenced by the viral strain. These aspects are relevant and require further investigations considering that during the recent outbreak of JEV in Australia in 2021/2022 over 80 piggeries were affected [84]. Interestingly, this outbreak was caused by a recently emerged genotype IV virus [85], which had a high mutation rate, possibly enabling a fast virus evolution and adaptation [86]. To our knowledge it is not known whether direct transmission events between pigs took place in Australia and whether the characteristics of the virus changed during the epidemic. We recommend that this should be investigated.

We also aimed to identify a possible change of the *in vitro* phenotype caused by the *in vivo* passaging by comparing the viral growth kinetics in insect and porcine cells. The comparison of P0 with P10 demonstrated much faster growth of P0, which pointed to a cell culture adaptation phenomenon. Indeed, P0 had been cultured for three passages to create the master and working stock termed "P0". Rapid cell culture adaptation effects are well-known for many viruses. For instance, only three *in vitro* passages of WNV increase replication characteristics [87]. For JEV, five passages in cell culture resulted in increased glycosaminoglycan receptor binding [88,89]. Therefore, we also compared the growth kinetics of JEV from P1, P5 and P10, which were all isolated by antibody-dependent enhancement on macrophages. Also, this comparison did not reveal the expected adaptation to porcine cells, indicating that our *in vitro* models may not be suitable to address species adaptations associated with fitness changes.

After only one passage we found a jump in genetic diversity indicating a rapid replacement of dominant populations present in P0 by minority variants. One could speculate that this effect may be related to a dramatic reduction in cell-culture adapted variants explaining the delayed *in vitro* replication of pig-passaged JEV. Nevertheless, no prominent mutation on the E protein was identified that could have explained this effect. Therefore, it could simply be related to the rapid ability of Flaviviruses to recover lost diversity following bottleneck effects (in this case the cell culture) [62].

Interestingly, from P1 to P10 the genetic diversity was stable indicating a balance between mutational diversification and selection processes. This observation may relate to the fact that JEV is well-adapted to pigs. However, $F_{ST}$ steadily increased until P5-6 pointing to selection of existing variants and/or new mutations within the first passages of the animal experiment. To investigate the selection of viral variants in detail, we analyzed the trajectories of all nucleotide variations reaching at least a frequency of 35% during the passages. Some of these mutations were classified as belonging to seven different haplotypes that contained at least two mutations with quasi-identical frequencies over the passages and were therefore most likely present on a single viral genome. Some of these haplotypes were already detected in the P0 swarm and were therefore clearly selected or deselected during passages. In particular, haplotype 2a and 2b

have increased fitness in pigs as they were selected in the two surviving lines B and C. In contrast, haplotype 3 was selected only in line A and reached nearly 100% before extinction of this line at P4. Of note, this haplotype as well as haplotype 4 did not contain mutations resulting in amino acid changes indicating the importance of other elements such as RNA secondary structures or codon optimization during selection.

As haplotypes 4–7 were not detected in P0, we cannot definitively conclude if they represent rare minor variants or if they were generated by simultaneous mutations on one genome. Most likely, haplotype 7 emerged by mutations as it was first detected after P5. In addition to haplotype 7, five single mutations arose during passaging. This is in line with the fact that random mutations will in most cases be deleterious for a virus [63].

Overall, six nucleotide deviations led to amino acid changes. In line C, we found a mutation within prM at position 515 of the genome, in which a polar threonine was replaced by a non-polar isoleucine (prM T13I). This was not described in a published JEV genome so far, although this position is not particularly conserved between different Flaviviruses [90]. Studies on ZIKV and JEV showed that point mutations in the prM protein can alter the infectivity of the virus [91–93]. It should be noted that in the competition experiment with the simultaneous infection with line B and C, prM T13I had a reduced frequency.

Only two amino acid changes were found in the E protein at positions 2074 (amino acid (AA) position 366) and 2098 (AA374). The S366P change was on haplotype 1, which was de-selected. M374L is more interesting as it emerged only at P7 in line B and was positively selected in the competition experiment. We were only able to find 374L in West Nile viruses [94,95]. It is located in the DIII domain of the E protein, which is partly exposed and targeted by antibodies. Interestingly, AA374 is the only variable position in this conserved domain (AA373-379) [96].

An amino acid change in the genomic position 6356 was localized in the NS3 protein at AA583 changing valine to alanine. This change was only selected in P1 of line B and then de-selected. 583A was not present in any of the JEV sequence data published on NCBI but found to be variable when comparing different Flaviviruses. While Dengue viruses and Zika viruses often harbor a tryptophan, Usutu virus has a valine and Saint Louis encephalitis virus a phenyl-alanine at this position [97–100]. In fact, the 583 position does not localize to the conserved RNA helicase domain [101,102].

At the genomic position 8500 within the NS5 gene, the amino acid change N275D was strongly selected in the pre-existing haplotypes 2a and 2b in lines B and C. N275D was described in two JEV isolates (GenBank: MH753129.1, JN381843.1) and other Flaviviruses like West Nile virus, yellow fever virus, Yokose virus and Murray valley encephalitis virus [103–105]. AA275 is located in a linker site between the functional domains of the RNA-dependent RNA polymerase and the methyltransferase [105]. The linker spans the amino acids 266 to 275 and shows little conservations between Flaviviruses. Removal of the linker does not abrogate polymerase activity [106]. Since the N275D variant was efficiently selected in both lines in pigs, it would be interesting to investigate this position in future studies.

The final amino acid change was caused by a mutation at genomic position 9658 first detected in P7 of line B. It resulted in a change of threonine to alanine in the NS5 protein (T661A), previously found in JEV (GenBank AB830335), replacing a polar with a non-polar amino acid at the end of a predicted alpha helix. While threonine can bend the alpha helix, ala-nine represents a stabilizing amino acid [107]. The alpha helix flanks a conserved motif of the NS5 protein [108]. The aspartate residues 667 and 668 of this conserved motif are hypothe-sized to play a role in the regulation of magnesium cations during the polymerization [109]. Although T661A was selected in line B it did not have a competitive advantage in the lines B/C co-infection.

Our study had several limitations. Our facilities did not permit to include a control group with alternating cycling between mosquitos and pigs, to be certain that all selections observed are linked to single host cycling in pigs. Artificial infection of mosquitos by blood feeding typically requires higher virus concentrations than those present in the serum of pigs and would therefore have required a cell-culture amplification step. We opted against this due to the danger of impacting the selection pressure. For these reasons, we decided to use only initial stock as comparison for possible adaptations. The selection of pre-existing haplotypes and the relatively high genomic stability suggested that using the initial JEV stock is valid after all. Another limitation was that the passaging was performed with serum and not with swabs samples. The reason for selecting the serum was to have a source of virus with sufficiently high titers to infect pigs efficiently during the animal experiment. We do not expect a major impact from this choice, considering that the limited comparative genomic analysis of swab and serum sequences presented in this manuscript does not support major differences.

Taken together, passaging JEV in pigs to mimic a series of direct transmission can increase its fitness in pigs in terms of higher viremia and increased mucosal shedding. Nevertheless, the pig's immune system remains effective in controlling the infection and the overall disease characteristics remain similar. Passaging is associated with both a dominant selection of pre-existing haplotypes as well as *de novo* mutations. Despite these strong selection processes the overall genetic diversity is maintained during passaging. Studies are ongoing to determine if the described viral adaptation results in a loss of fitness in mosquitoes. Our work contributes to understanding how Flaviviruses maintain fitness after interruption of alternating host cycling. Despite strong genetic selection processes following passaging in pigs, the impact on disease progression, viral tropism and immune response characteristics were limited, pointing to a high level of JEV adaptation to the pig host. Fortunately, JEV did not appear to adapt for increased direct transmission.

## Supporting information

**S1 Text. Supplementary Methods and References.**
(PDF)

**S1 Fig. Tissue distribution of JEV RNA during passaging.**
(PDF)

**S2 Fig. Histopathological analyses of CNS from JEV infected pigs.**
(PDF)

**S3 Fig. Unmodulated or undetectable cytokines in the serum of JEV-infected pigs.**
(PDF)

**S4 Fig. Nucleotide diversity $\pi$ and fixation index $F_{ST}$ for individual viral genes and UTRs.**
(PDF)

**S1 Table. Age of piglets at the day of infection.**
(XLSX)

**S2 Table. Number of reads and the covering depth of NGS samples.**
(XLSX)

**S3 Table. Linear regression and Dunnett's multiple comparison test of in vitro growth curves (Fig 2 data).**
(XLSX)

**S4 Table. Viral mutations and frequencies within each passage.**
(XLSX)

**S5 Table. Nucleotide diversity π and fixation index $F_{ST}$ for each passage.**
(XLSX)

**S6 Table. Comparative frequencies of major selected SNVs in serum and swab samples.**
(XLSX)

**S7 Table. Comparative frequencies of all SNVs in serum and swab samples for pig 2029.**
(XLSX)

**S1 Data. File containing data points of Figs 1, 2, 3, 4, and 6.**
(PZFX)

## Acknowledgments

We thank Katarzyna Sliz and Daniel Brechbühl for the animal care during the experiments and Urs Pauli and Ruth Knorr for the support regarding biosafety. We are grateful to Razieh Ardali for her support in the analysis of the transcriptomic data. We greatly appreciated all the help and work of Pamela Nicholson, Daniela Steiner and Samia Imadjane from the Next Generation Sequencing Platform (University of Bern, Bern, Switzerland) for the BRB-sequencing and the total RNA-sequencing. We thank Remi Charrel and Antoine Nougairede (Aix-Marseille Université, Marseille, France) for providing the JEV strain Laos and Dr. Jörg Seebach (University of Geneva, Switzerland) for the PEDSV.15 cell line.

## Author Contributions

**Conceptualization:** Nicolas Ruggli, Obdulio García-Nicolás, Artur Summerfield.

**Data curation:** Andrea Marti, Alexander Nater, Francisco Brito.

**Formal analysis:** Andrea Marti, Alexander Nater, Francisco Brito.

**Funding acquisition:** Artur Summerfield.

**Investigation:** Andrea Marti, Jenny Pego Magalhaes, Lea Almeida, Marta Lewandowska, Llorenç Grau-Roma, Obdulio García-Nicolás.

**Methodology:** Andrea Marti, Alexander Nater, Jenny Pego Magalhaes, Lea Almeida, Marta Lewandowska, Matthias Liniger, Llorenç Grau-Roma, Obdulio García-Nicolás.

**Project administration:** Andrea Marti, Nicolas Ruggli, Obdulio García-Nicolás, Artur Summerfield.

**Resources:** Matthias Liniger, Nicolas Ruggli, Artur Summerfield.

**Software:** Alexander Nater, Fadi G. Alnaji, Marco Vignuzzi.

**Supervision:** Obdulio García-Nicolás, Artur Summerfield.

**Validation:** Obdulio García-Nicolás, Artur Summerfield.

**Visualization:** Andrea Marti, Francisco Brito, Artur Summerfield.

**Writing – original draft:** Andrea Marti, Artur Summerfield.

**Writing – review & editing:** Alexander Nater, Marta Lewandowska, Matthias Liniger, Nicolas Ruggli, Llorenç Grau-Roma, Fadi G. Alnaji, Marco Vignuzzi, Obdulio García-Nicolás.

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
