## [Decision Letter · Decision Letter 0]

22 Apr 2024

Dear Professor Summerfield,

Thank you very much for submitting your manuscript "Fitness adaptations of Japanese encephalitis virus in pigs following vector-free serial passaging" for consideration at PLOS Pathogens. As with all papers reviewed by the journal, your manuscript was reviewed by members of the editorial board and by several independent reviewers. In light of the reviews (below this email), we would like to invite the resubmission of a significantly-revised version that takes into account the reviewers' comments.

We cannot make any decision about publication until we have seen the revised manuscript and your response to the reviewers' comments. Your revised manuscript is also likely to be sent to reviewers for further evaluation.

Sincerely,

Michael Letko, PhD

Section Editor

PLOS Pathogens

Michael Letko

Section Editor

PLOS Pathogens

Michael Malim

Editor-in-Chief

PLOS Pathogens

orcid.org/0000-0002-7699-2064

Reviewer's Responses to Questions

**Part I - Summary**

Reviewer #1: This is the manuscript from the group that discovered and published the vector-free JEV transmission in pigs. The authors build on their expertise and further extended studies to serial passaging. The experimental design partially reproduces the possible non-vector transmissions on JEV-affected swine herds in endemic countries outside the mosquito season and models potential transmission events after introduction to naïve herds. The authors demonstrated that after serial passaging in pigs, JEV induced enchased clinical signs in some experimental conditions and altered immune responses. Using NGS, the emergence of new JEV variants was also identified.

The authors applied a well-established toolbox for JEV research and NGS, providing interesting results. The study was conducted in the native amplifying host—pigs. Counting that JEV is a BSL3 pathogen, this is a unique study because only a small number of research groups can conduct studies with BSL3 pathogens at such level in the native hosts, particularly large agricultural animals.

Reviewer #2: This research team previously demonstrated that the Japanese encephalitis virus (JEV) can undergo vector-free transmission among experimental pigs. This manuscript explores the changes in viral phenotype and molecular level after 10 passages of JEV transmission among pigs through direct (vector-free) transmissions. While the innovative research objective is appealing, the results lack persuasiveness, possibly due to the influence of pig experiments and specimen analysis methods, as well as insufficient in-depth discussion of the research findings.

Reviewer #3: See attached

**Part II – Major Issues: Key Experiments Required for Acceptance**

Reviewer #1: (No Response)

Reviewer #2: General comments:

1. When exploring the impact of transmission between a single vector or vertebrate host, or between dual hosts, on the evolution of arboviruses, three sets of experiments should be conducted: virus infection only in the vector, virus infection only in the vertebrate host, and circulation infection between the dual hosts. In this study, the experiment only investigated JEV infection among pigs, without conducting the other two sets of experiments (at least the circulation infection between the vector and pigs should be conducted, as the probability of vertical transmission of JEV between vectors is very low). Therefore, it is difficult to infer that the observed changes in JEV phenotype and genotype are solely due to direct transmission of JEV among pigs.

2. This research team previously demonstrated through animal experiments that the Japanese encephalitis virus (JEV) can undergo direct transmission among pigs via the oro-nanal route. However, in this manuscript, when conducting consecutive infections of JEV among pigs via direct transmission, they utilized the virus from the serum of previously infected pigs to infect the next batch of pigs (via the intranasal route), which could very likely introduce bias into the research results.

3. This manuscript only presented JEV in pig samples by quantifying viral RNA and did not present results regarding infectious virus. Particularly, the rapid rise of neutralizing antibodies in serum and the presence of substances in oro-nasal secretions that can degrade viral material suggest that measuring viral RNA alone may not exclude the influence of these factors.

4. When arboviruses infect vectors or vertebrate hosts, there is a phenomenon known as tissue-associated selection. However, in this manuscript, the analysis of the effects of direct transmission of JEV among pigs on the molecular level only focused on the virus in serum, which may not accurately reveal the actual situation.

5. The authors also express doubts about the significance of direct transmission of JEV among pigs in field settings. In Figure 6 and their previously published paper, they demonstrate poor effectiveness of direct transmission. Additionally, some literature indicates that JEV viral RNA can be detected in oro-nanal secretions collected from pigs in the field, but infectious virus cannot be detected. Therefore, the attempt in this paper to conduct 10 passages of JEV transmission among pigs through direct (vector-free) transmission seems somewhat unrealistic.

6. This manuscript presents numerous research results, but most of the segments provide descriptions and explanations without integrating the results into a more comprehensive interpretation and discussion. This is indeed regrettable.

Reviewer #3: See attached

**Part III – Minor Issues: Editorial and Data Presentation Modifications**

Reviewer #1: MATERIALS AND METHODS

Line 131-132: Please provide more details on how pig infection was conducted or provide references.

“Swabs and serum samples were taken at 0, 3 and 4 days post-infection (dpi)” – Please clarify when exactly day 0 swabs were taken, before or after infection, how deep swabs were taken, and what was done with swabs immediately after collection (frozen directly or diluted, if diluted then how, etc.).

“The day 4 serum was used for infection of three new pigs intranasally, keeping the lines separated (A, B and C).” – Please provide the volume of the sera for infection.

From the provided description, it is unclear what “lines” and “A, B, and C” are. Explanations are provided in the text below, but now it is confusing. Please change the language for clarity.

Line 213: “P0 sample …” – it is not clear what is P0. Is it the JEV inoculum stock?

If P0 is JEV inoculum stock, please go through the entire text, figures, and figure legends, and change P0 to the “inoculum” or “initial JEV stock”, or something else because P0 in the given context is confusing and misleading.

S8 Methods – more details are needed on how JEV isolated from serum was titrated for subsequent growth curves.

RESULT

Line 242: Lines – A, B, C – should be shown on Fig 1a to easier refer text to the figure.

Line 247: “A statistical comparison of P0” – do you mean P1? There is no P0 in Fig 1a.

Lines 254-256: “It is important to note that clinical differences may also have been influenced by age effects. The pigs used for the data shown in Fig 1 and 2 were from different litters. these reasons we further 257 characterized the viruses isolated from P1-P10.” – It is not clear what the authors mean here: “…. pigs used for the data shown in Fig 1 and 2,” does it mean that in addition to the JEV passage represented in Fig 1, additional animal experiments were conducted specifically for data conducted in Fig 2? If yes, these additional experiments should be described in MM. If not, please edit the text to clearly deliver information that Fig 1 and 2 represent data from the same animal experiment, and that pigs used in different passages (P1, P2, etc., Fig 1) have different ages.

The age of pigs in each passage should be clearly described in MM.

Fig 2 - the title, figure, and legend should clearly describe the sample type. It is unclear whether these are serum, nasal swab, or tissue samples.

Line 291: (Fig 3b-h) – “h” should be removed.

Line 300: Fig 2h – change to Fig3h.

Line 359: Please edit the section title.

Line 361: “In P0, 220 nucleotide deviations from the consensus Laos sequence were found.” – Here, the authors compare the inoculum stock sequences to the reference sequence available in NCBI (GenBank: KC196115.1); is it correct? Please clarify and edit the text for clarity.

Line 362: “During passaging, additional 8% of the nucleotide positions mutated.” – Here, NGS data from comparing in vivo JEV sequences as referenced to the de novo in-house generated inoculum JEV stock-specific sequences are described; is it correct? Please clarify and edit the text.

Reviewer #2: Specific comments:

1. Line 25 and 68: The role of birds in the transmission of Japanese encephalitis virus (JEV) remains unclear because current evidence mostly consists of indirect evidence, including antibody positivity rates and results from animal experiments. If birds do indeed serve as natural reservoirs, it raises the question of why there are not many JEV gene sequences isolated from birds in GenBank.

2. Line 26: Currently, vector-free transmission of JEV has only been observed in experimental pigs and has not yet been confirmed in the field. It is suggested to rephrase it as "...observed in experimental pigs."

3. Line 132: "The day 4 serum was used for infection of three new pigs intranasally...". Explanation is needed as to why serum virus was used instead of virus from oro-nasal secretions and why the intranasal route was chosen over the oro-nasal route. Additionally, the inoculation dose (ul or ml/animal) should be specified. Here, when referring to "serum," it should be clarified how it was prepared after blood collection. Was it allowed to clot, or was an anticoagulant added? Typically, the serum is obtained by centrifugation after blood clotting, but the clotting process can significantly reduce the amount of infectious virus, making it less suitable for samples with low viremia, including JEV.

4. Line 184 and Fig 2: It should be explained why both mosquito C6/36 cells and pig PEDSV.15 cells were used for JEV virus titration.

5. Line 190: The S7 methods section does not describe the TCID50 measurement method but rather the Focus Forming Units (FFU) measurement method. Throughout the manuscript, the quantification of infectious virus is stated in terms of TCID50/ml. Please clarify which method is used. Typically, C6/36 cells do not die after JEV infection. How was the TCID50 quantification performed?

6. Lines 197-198: This manuscript only specifies that viral RNA used for viral genome analyses is derived from serum or swab samples. Figure 7 does not specify the source of viral RNA, while Figures 8 and 9 indicate that viral RNA is from serum. The source of viral RNA could significantly impact the obtained results, so it should be clearly stated. If analyzing viral RNA from a single tissue or organ source, the reasons for this choice should be explained, along with a discussion of potential limitations.

7. Line 250 and others: Regarding the units for viral RNA quantification, throughout the manuscript and figures, it is stated as viral copies/μl serum, with values reaching up to ~105, indicating up to ~108 per ml. This value is remarkably high for samples from JEV-infected pigs, so please confirm if this is accurate. In their 2016 article published in Nature Communications, the authors used relative quantitative RT-qPCR, defining 1 TCID50 as 1 RNA unit. In this manuscript, absolute quantitative RT-qPCR is used. Could authors provide relevant information to facilitate the estimation of the amount of infectious virus?

8. Line 255: The authors mention that the clinical presentation of pigs may be influenced by their age; however, experimental pig ages are not provided in the text, especially concerning the ages of pigs in different passages.

9. Lines 273-279: It should be explained why all serum viruses need to be rescued on macrophages before conducting the replication capacity comparison. Additionally, the impact of this additional step on in vitro experiments should be discussed and addressed. Utilizing infectious virus directly from serum, rather than the rescued virus, for in vitro experiments would more accurately reflect real-world conditions. The results obtained from in vitro experiments are not compared and discussed alongside in vivo results and viral genome analysis results.

10. Lines 139 and 283: In the in vivo experiment presented in Fig 3, P10B and P10C were mixed in a 1:1 ratio. However, the in vitro results shown in Fig 2 indicate that the viruses from line B and line C are different. It may not be suitable to mix them for subsequent experiments. The experimental challenges and potential impacts on the results should be explained and discussed.

11. Line 312: Analysis of cytokine profiles revealed that JEV infection of pigs by intranasal route induces a Th1 response, but this finding is not compared, explained, or discussed alongside the subsequent Gene Set Enrichment Analysis (GSEA) results. Furthermore, there is no discussion comparing these results with those from other JEV animal models, infection routes, or analyses of clinical samples from humans.

12. Lines 315-317 and Fig 4i: This experiment found that regardless of the experimental group, the time and intensity of neutralizing antibody responses were the same. However, Fig 3C shows differences in clinical presentation. How can this be explained? What role might cytokines or T cell responses play?

13. Line 352 and Fig 6: The experimental results indicate that direct transmission via the oro-nasal route is less effective. The authors attribute this to the density of housing, but cannot rule out the possibility that it is due to the lower quantity of infectious virus present in oro-nasal secretions. In Fig 1D and 1E, it is speculated that the quantity of infectious virus in serum is much higher than in oro-nasal secretions. Therefore, using serum virus for 10 passages of direct transmission may significantly affect the bottleneck effect, thereby rendering the results of this manuscript unable to reflect reality (as mentioned in general comment 2). While the results in Fig 6 show no difference in direct transmission between P1* and P11 viruses, the results in Fig 3E indicate differences in nasal shedding. How can this be explained?

14. Line 359 and Fig 7: The authors did not provide basic information regarding the NGS analysis, such as the number of reads, the proportion of reads belonging to JEV, and the resulting coverage depth across the viral genome. Why were reads filtered for a frequency above 35%?

The results obtained from Fig 7b and 7c should be integrated, explained, and discussed alongside the results from Fig 1, 2, and 3.

15. Line 415 and Fig 9: P10B and P10C were mixed at a ratio of 1:1 based on the infectious virus (line 139: forming 1.8X105 TCID50), which does not necessarily reflect a 1:1 ratio at the viral genome level. The authors should explain and discuss the potential impact of this discrepancy.

16. Comparative analysis of results from Fig 1 and 3 requires explanation and discussion, highlighting three points: (1) Why is the viral RNA quantity in Fig 3E and 3F significantly higher than in Fig 1E? (2) The viral RNA quantity in Fig 3D, 3E, and 3F is similar, but why is it noticeably higher in Fig 1D than in Fig 1E? (3) While the overall trend in Fig 1C is downward, why is P11 higher than P1* in Fig 3C?

Reviewer #3: See attached

PLOS authors have the option to publish the peer review history of their article (what does this mean?). If published, this will include your full peer review and any attached files.

Reviewer #1: **Yes: **Uladzimir Karniychuk

Reviewer #2: No

Reviewer #3: No
---

## [Decision Letter · Decision Letter 1]

10 Jul 2024

Dear Professor Summerfield,

Thank you very much for submitting your manuscript "Fitness adaptations of Japanese encephalitis virus in pigs following vector-free serial passaging" for consideration at PLOS Pathogens. As with all papers reviewed by the journal, your manuscript was reviewed by members of the editorial board and by several independent reviewers. The reviewers appreciated the attention to an important topic. Based on the reviews, we are likely to accept this manuscript for publication, providing that you modify the manuscript according to the review recommendations.

Sincerely,

Scott P. Kenney, Ph.D.

Guest Editor

PLOS Pathogens

Michael Letko

Section Editor

PLOS Pathogens

Michael Malim

Editor-in-Chief

PLOS Pathogens

orcid.org/0000-0002-7699-2064

Reviewer Comments (if any, and for reference):

Reviewer's Responses to Questions

**Part I - Summary**

Reviewer #2: The revised manuscript has undergone significant improvement. Despite the authors' inability to enhance the experimental design or conduct further experiments due to practical constraints, they have included additional explanations in the discussion concerning potential limitations that could affect the study. Therefore, while the authors' response is less than ideal, it is deemed acceptable.

1. Although conducting experiments involving infection between the vector and pigs was impractical for logistical reasons, conducting virus infection experiments solely in the vector (which is comparatively simpler) would likely enhance the interpretation of the research findings.

2. Regarding concerns about potential bias arising from using virus extracted from the serum of previously infected pigs to infect the subsequent batch of pigs, the authors were fortunate that the NGS results from serum and swab samples in this study were similar (Table S6 and S7), thus partially addressing the aforementioned concerns.

Reviewer #3: As per previous review

**Part II – Major Issues: Key Experiments Required for Acceptance**

Reviewer #2: No further comment.

Reviewer #3: NA

**Part III – Minor Issues: Editorial and Data Presentation Modifications**

Reviewer #2: No further comment.

Reviewer #3: Introduction

In the previous comments I requested to refer to mosquitoes as vectors not hosts to be clear what you are referring to, but this remains the case on line 64 and on line 66. This is confusing because JEV can circulate in multiple host species. Please be clear in the text you mean host and vectors not two hosts.

Paragraph on Line 80: You need to better explain why DT would be less likely than mosquito-host transmission when there are more resistant individuals in the population compared to a fully susceptible population. In addition, line 80 specifically is poorly phrased – you talk about DTs being rare in endemic areas but then give evidence for this based on piglets in Cambodia. It would be better to start this section by explaining the importance of resistance generally – then giving examples of where resistance comes from (maternal antibodies, vaccine, exposure).

Methods

In the previous review, I requested more detail as to the statistics carried out in the methods, but there now seems to be even less! I recommend fully writing out the statistical analyses for each hypothesis to be tested within the methods. This helps the reader to follow what you have done and why. This is particularly important in a study such as this which has a lot of components. I understand that you have put the stats in the figure legends - but you wouldn't think to do that for other parts of the methods, so why the statistical analysis methodology?

Results

Please omit the use of ‘interestingly’ and ‘surprisingly’ in the results

There is some repetition on lines 286 and 299

PLOS authors have the option to publish the peer review history of their article (what does this mean?). If published, this will include your full peer review and any attached files.

Reviewer #2: No

Reviewer #3: No

Figure Files:

Data Requirements:

Reproducibility:

References:

---

## [Editor Report · Decision Letter 2]

2 Aug 2024

Dear Professor Summerfield,

We are pleased to inform you that your manuscript 'Fitness adaptations of Japanese encephalitis virus in pigs following vector-free serial passaging' has been provisionally accepted for publication in PLOS Pathogens.

Best regards,

Scott P. Kenney, Ph.D.

Guest Editor

PLOS Pathogens

Michael Letko

Section Editor

PLOS Pathogens

Michael Malim

Editor-in-Chief

PLOS Pathogens

orcid.org/0000-0002-7699-2064
---

## [Editor Report · Acceptance letter]

20 Aug 2024

Dear Professor Summerfield,

We are delighted to inform you that your manuscript, "Fitness adaptations of Japanese encephalitis virus in pigs following vector-free serial passaging," has been formally accepted for publication in PLOS Pathogens.

Best regards,

Michael Malim

Editor-in-Chief

PLOS Pathogens

orcid.org/0000-0002-7699-2064